# MEMAGENT: Reshaping Long-Context LLM with Multi-Conv RL-based Memory Agent

**Hongli Yu**[1,2,3]   **Tinghong Chen**[1]   **Jiangtao Feng**[1]   **Jiangjie Chen**[2,3]   **Weinan Dai**[1,2,3]

**Qiying Yu**[1,2,3]   **Ya-Qin Zhang**[1,3]   **Wei-Ying Ma**[1,3]   **Jingjing Liu**[1,3]   **Mingxuan Wang**[2,3*]

**Hao Zhou**[1,3*]
[1] Institute for AI Industry Research (AIR), Tsinghua University
[2] ByteDance Seed
[3] SIA-Lab of Tsinghua AIR and ByteDance Seed

## ABSTRACT

Despite improvements by length extrapolation, efficient attention and memory modules, handling infinitely long documents without performance degradation during extrapolation remains the ultimate challenge in long-text processing. To solve this problem, we introduce a novel agent workflow, MEMAGENT, which processes text in segments and updates memory through an overwrite strategy, addressing the challenge of long-context task through enhanced memory management. We further extend the DAPO algorithm to directly optimize memory ability in an end-to-end fashion, facilitating training via independent-context multi-conversation generation. Experimental results demonstrate that MEMAGENT has superb long-context capabilities, being able to extrapolate from an 8K context to a 3.5M QA task with a performance loss of less than 10% and achieving over 95% on the 512K NIAH test.

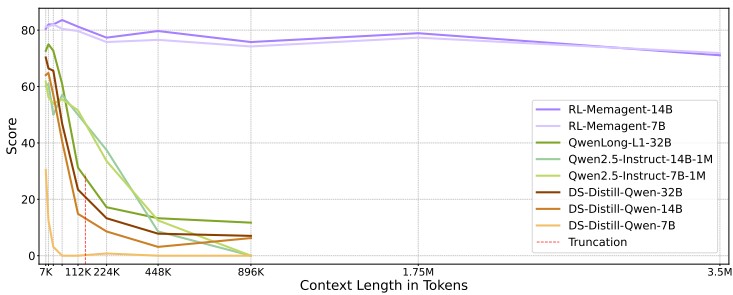

Figure 1: Accuracy scores of RULER-HQA (Hsieh et al., 2024; Yang et al., 2018) . Even models that employ long-context continual pretraining and extrapolation techniques fail to maintain consistent performance. In contrast, MEMAGENT with RL only demonstrates marginal performance dropping.

## 1 INTRODUCTION

While having demonstrated impressive capabilities (OpenAI, 2024; DeepMind, 2024; XAI, 2024; Anthropic, 2024; OpenAI, 2023), industry-level Large Language Model (LLM) systems (Anthropic, 2025; Li et al., 2025a; Liu et al., 2024; Yen et al., 2024) still face a critical challenge: how to handle long contexts effectively - processing an entire book, executing a complex chain of reasoning over many steps, or managing the long-term memory of an agent system - all these complex tasks can generate overflowing text that quickly explodes the typical-size context window of current LLMs.

---

*Corresponding author.

Existing approaches to long-context tasks are three-pronged. The first involves length extrapolation methods by shifting the positional embeddings in order to extend the context window of the model (Su et al., 2024; bloc97, 2023; Chen et al., 2023; Peng et al., 2023b; An et al., 2024), plus continued pre-training (Liu et al., 2023; Xiong et al., 2023; Gao et al., 2025). Despite promising potential, these methods often suffer from performance degradation and slow processing speed due to $O(n^2)$ computational complexity when applied to extremely long text. The second school of methods leverages sparse attention (Beltagy et al., 2020; Zhao et al., 2019; Xiao et al., 2023) and linear attention mechanisms (Child et al., 2019; Katharopoulos et al., 2020) to reduce the complexity of attention for more efficient processing of longer sequences. However, this typically requires training from scratch, with inherent adversities such as linear attention facing difficulties in parallel training or sparse attention depending on human-defined patterns. The last line of inquiry investigates context compression (Jiang et al., 2023; Li et al., 2023; Behrouz et al., 2024; Zhang et al., 2024), which aims to condense information in token-level or external-memory-plugin modules. Such approaches often struggle with extrapolation, and require the integration of additional modules or context operations, which ineluctably disrupts the standard generation process and hinders compatibility as well as parallelization.

Hence, a successful LLM with strong long-context capabilities requires the trinity of: 1) processing infinite length of text; 2) scaling without significant performance drop; and 3) efficient decoding with linear complexity. To pursue this quest, we return to the basic intuition behind long-context modeling (Miller et al., 1956; Hochreiter & Schmidhuber, 1997; Graves et al., 2014; Weston et al., 2014). When humans process long-context information, we tend to abstract out the main revealing conceptions to capture the essence of the whole text, often by making notes of critical details or using short-handed stenograph to record the key points, while discarding redundant and irrelevant data. We do not attempt to memorize every single fact or each small piece of information; instead, we focus our intellectual energy on more important aspects of the task at hand. This selective attention not only simplifies the process but also aids in tackling complex problems more efficiently.

Following this anthropocentric intuition, we propose a novel use of Reinforcement Learning (RL) to equip LLMs with a dynamically updated fixed-length 'memory', as illustrated in Figure 2. During inference, the LLM processes the input text segment-by-segment. As it reads each segment, the model proactively and selectively updates the memory, which then contributes to the generation of the final output after all relevant messages are aggregated and synergized in the memory. This clever mechanism allows the LLM to flexibly handle arbitrary text lengths while maintaining a linear time complexity during processing, since the length of the memory is fixed, which leads to a fixed context window size for the model. This segment-based approach generates multiple outputs from a single long-text input, requiring multiple rounds of memory updates and a final round for the generation of the final response. Training this type of agent workflow, which enables dialogues across multiple independent contexts, is still an unexplored territory in current LLM study. Existing systems typically handle workflow trajectories via alternating tool calls or environment feedback by either simply concatenating (Ouyang et al., 2025; Jin et al., 2025) them or using a sliding window (Feng et al., 2025) approach, which lacks flexibility and scalability in practice. Our MEMAGENT approach, instead, proposes that treats each context-independent conversation as an optimization objective. Based on the DAPO (Yu et al., 2025) algorithm, we implement the Multi-Conv DAPO to optimize an arbitrary agent workflow by verifiable outcome reward.

In our experiments, an RL-trained model with a modest 8K context window (with a 1024-token memory and a 5000-token document chunk) trained on 60K length documents exhibits consistently superb capabilities for Question Answering (QA) tasks on documents of up to 3.5 million tokens, without performance drop and with linear computation cost. This demonstratively showcases the efficiency and scalability of our long-context memory approach.

Our major contributions are threefold:

- We introduce a novel approach that enables LLMs to process arbitrarily long inputs within limited context window under linear time complexity during inference, overcoming a significant bottleneck in long-context processing.
- We design an agent workflow to implement this mechanism and propose an end-to-end training approach using the multi-conversation DAPO algorithm.

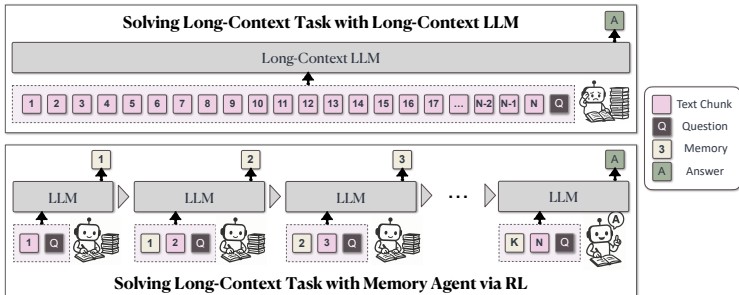

Figure 2: MEMAGENT is inspired by the way humans process long documents. It divides the document into multiple chunks and allows LLMs to process them iteratively, recording relevant information in memory. Finally, LLMs generate answers based on the information stored in the memory.

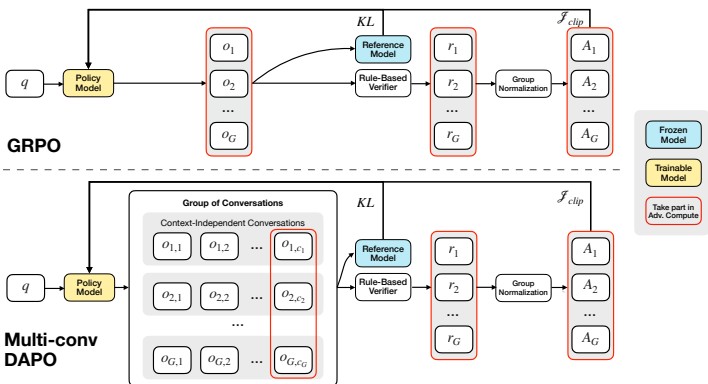

Figure 3: Comparison between vanilla GRPO and Multi-Conv DAPO. During the rollout phase of Multi-conv DAPO, each sample generates multiple conversations. The answer contained in the final conversation is used to compute the reward and advantage, which are then employed to optimize all preceding conversations.

- We empirically demonstrate that our RL-trained method allows models to extrapolate to vastly long documents with minimal performance degradation, pushing the boundaries of what is currently achievable in long-context LLM systems.

## 2 METHODOLOGY

In this section, we describe the details of MEMAGENT approach for solving long-context tasks, including the overall workflow (§ 2.1), Multi-conv RL algorithm for training MEMAGENT (§ 2.2) and the formal modeling of our architecture(§ 2.3).

### 2.1 THE MEMAGENT WORKFLOW: RL-SHAPED MEMORY FOR UNBOUNDED CONTEXTS

As illustrated in Figure 2, MEMAGENT views an arbitrarily long document not as a monolithic block but as a controlled *stream* of evidence. At every step, the model sees exactly two things: the next chunk of text and a compact, fixed-length *memory* that summarizes everything deemed important so far. Crucially, the memory is just a sequence of ordinary tokens inside the context window, so the core generation process of the base LLM remains unchanged.

After reading a new chunk, the model overwrites the previous memory with an updated one. This **overwrite** strategy seems almost too simple, yet it is precisely what enables the system to scale: because memory length never grows, the total compute per chunk stays $O(1)$ and end-to-end complexity is strictly linear to the number of chunks. We formulate the overwrite decision as a reinforcement

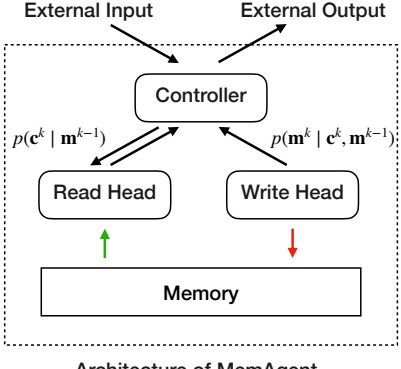 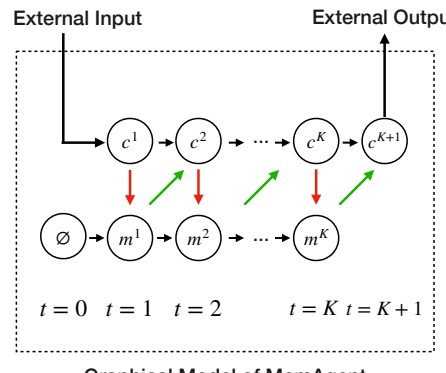

Figure 4: The architecture and graphic model of MEMAGENT. The memory is modeled as a latent memory variable, thereby enabling the decomposition of the autoregressive language model into multiple steps of reading from and writing to the memory.

learning problem: the agent is rewarded for retaining information that will later prove useful and for discarding distractors that would waste precious tokens. By optimizing this objective with our newly introduced multi-conversation DAPO algorithm (detailed in § 2.2), the model learns to compress aggressively while preserving answer-critical facts.

The workflow naturally decomposes inference into two modules. Within the **Context-Processing** module the model iterates over chunks, updating memory with a prompt template (Table 5, top). Once the stream is exhausted, a final **Answer-Generation** module is invoked (Table 5, bottom) where the model consults only the problem statement and the memory to produce its boxed answer. Because positional embeddings are never re-scaled or patched, the same tokenization and attention layout apply in both modules, unlocking the model's latent length-extrapolation capability without any architectural modifications.

MEMAGENT therefore enjoys three benefits from this design: (1) **Unlimited length**: the document can be millions of tokens because it is processed as a stream; (2) **No performance cliff**: RL encourages the memory to retain exactly the information needed, yielding near-lossless extrapolation (Figure 1); (3) **Linear cost**: a constant window size implies decoding time and memory consumption grow linearly with input length ($O(N)$) (detailed in § B.) This renders a practical recipe for turning any moderately context-sized LLM into an efficient long-context reasoner with minimal engineering overhead.

## 2.2 TRAINING MEMAGENT WITH MULTI-CONV RL

By viewing memory update in context processing for answer-generation tasks as part of the policy to be optimized by RL, we adopt the RLVR recipe (OpenAI, 2024; Guo et al., 2025; Seed et al., 2025) to train MEMAGENT. We adopt DAPO (Yu et al., 2025), an efficient and effective algorithm refined from Group Relative Policy Optimization (GRPO) (Shao et al., 2024), as our training algorithm. Due to the nature of our MEMAGENT approach, which generates multiple context-independent conversations for a single query as illustrated in Figure 2, we treat each conversation as an independent optimization target. This approach necessitates an extension of the loss computation from the conventional `(group, token)` structure to a new `(group, conversation, token)` dimensionality, as shown in Figure 3.

Formally, the policy model $\pi_{\theta_{\text{old}}}$ samples a group of $G$ individual responses $\{o_{i,j}\}_{i=1}^{G}$ for an input $x$. Let $n_i$ denote the number of generated conversations $(o_{i,1}, o_{i,2}, ..., o_{i,n_i})$ for a given sample $(q_i, a_i)$. $o_{i,j}$ further decomposes into token-level outputs $(o_{i,j,1}, o_{i,j,2}, ..., o_{i,j,|o_{i,j}|})$. The advantage value is derived from the conversation that contains the final answer, then uniformly applied to all conversations originating from the same sample, as shown in Eq 1. Following Dr. GRPO (Liu et al.,

Table 1: Main experimental results comparing model performance across various context lengths. All values represent accuracy (%).

| Model | Length | | | | | | | | | |
|---|---|---|---|---|---|---|---|---|---|---|
| | 7K | 14K | 28K | 56K | 112K | 224K | 448K | 896K | 1.75M | 3.5M |
| QwenLong-L1-32B | 72.66 | 75.00 | 72.66 | 60.94 | 31.25 | 17.19 | 13.28 | 11.72 | N/A | N/A |
| Qwen2.5-Instruct-14B-1M | 60.16 | 60.94 | 50.00 | 57.03 | 50.00 | 37.50 | 8.59 | 0.00 | N/A | N/A |
| Qwen2.5-Instruct-7B-1M | 61.72 | 56.25 | 53.91 | 55.47 | 51.56 | 33.59 | 12.50 | 0.00 | N/A | N/A |
| DS-Distill-Qwen-32B | 70.31 | 66.41 | 65.62 | 46.88 | 23.44 | 13.28 | 7.81 | 7.03 | N/A | N/A |
| DS-Distill-Qwen-14B | 64.06 | 64.84 | 57.03 | 40.62 | 14.84 | 8.59 | 3.12 | 6.25 | N/A | N/A |
| DS-Distill-Qwen-7B | 30.47 | 12.50 | 3.12 | 0.00 | 0.00 | 0.78 | 0.00 | 0.00 | N/A | N/A |
| Qwen2.5-Instruct-32B | 69.53 | 64.84 | 60.16 | 51.56 | 44.53 | 21.88 | 14.06 | 7.03 | N/A | N/A |
| Qwen2.5-Instruct-14B | 75.00 | 67.19 | 57.03 | 54.69 | 44.53 | 21.88 | 10.94 | 2.34 | N/A | N/A |
| Qwen2.5-Instruct-7B | 52.34 | 57.03 | 51.56 | 44.53 | 32.81 | 13.28 | 6.25 | 1.56 | N/A | N/A |
| **RL-MEMAGENT-14B** | 80.47 | **82.03** | **82.03** | **83.59** | **81.25** | **77.34** | **79.69** | **75.78** | **78.91** | 71.09 |
| **RL-MEMAGENT-7B** | **81.25** | 81.25 | **82.03** | 80.47 | 79.69 | 75.78 | 76.56 | 74.22 | 77.34 | **71.88** |

2025), we do not devide the advantage by the standard deviation. Eq 2 describes our loss function.

$$\hat{A}_{i,j,t} = R_i - \text{mean}(\{R_i\}_{i=1}^{G}) \tag{1}$$

$$
\begin{aligned}
\mathcal{J}_{\text{DAPO}}(\theta) = \quad & \mathbb{E}_{(q,a)\sim\mathcal{D},\{o_{i,j}\}_{i=1}^{G}\sim\pi_{\theta_{\text{old}}}(\cdot|q,\,o_{i,j-1})} \\
& \left[ \frac{1}{\sum_{i=1}^{G}\sum_{j=1}^{n_i}|o_{i,j}|} \sum_{i=1}^{G}\sum_{j=1}^{n_i}\sum_{t=1}^{|o_{i,j}|} \left( \mathcal{C}_{i,j,t} - \beta D_{\text{KL}}(\pi_\theta||\pi_{\text{ref}}) \right) \right] \\
\text{where} \quad & \mathcal{C}_{i,j,t} = \min\left( r_{i,j,t}(\theta)\hat{A}_{i,j,t},\ \text{clip}\left( r_{i,j,t}(\theta), 1-\varepsilon_{low}, 1+\varepsilon_{high} \right)\hat{A}_{i,j,t} \right) \\
& r_{i,j,t}(\theta) = \frac{\pi_\theta(o_{i,j,t} \mid q, o_{i,j,<t})}{\pi_{\theta_{\text{old}}}(o_{i,j,t} \mid q, o_{i,j,<t})}.
\end{aligned}
\tag{2}
$$

Following the RLVR recipe (Guo et al., 2025; Jin et al., 2025; Yu et al., 2025), we train the model with a final outcome reward computed by a rule-based verifier:

$$R(\hat{y}, y) = \mathbf{1}_{\texttt{is\_equiv(y,\hat{y})}} \tag{3}$$

where $\hat{y}$ is the predicted answer while $y$ refers to the ground truth.

## 2.3 RETHINKING MEMAGENT FROM AUTOREGRESSIVE MODELING PERSPECTIVES

Tto get a deeper sense of the MEMAGENT design, we propose to re-think language-model factorization in the following fashion. A standard autoregressive LLM factorizes the joint likelihood of a sequence $\mathbf{x}_{1:N}$ as $p(\mathbf{x}_{1:N}) = \prod_{n=1}^{N} p(x_n \mid \mathbf{x}_{1:n-1})$, implicitly assuming that every past token (or at least its hidden state) must stay in the active context. This is what turns quadratic attention into the long-context bottleneck.

MEMAGENT replaces the unbounded history with a fixed-length *memory* $\mathbf{m} \in \mathbb{V}^{M}$, as shown in Figure 4. The input text is streamed through the model in $K$ contiguous chunks $\mathbf{c}^{1}, \ldots, \mathbf{c}^{K}$ (each of length $\leq C$). After chunk $k$ is read, the model overwrites the panel with a new vector $\mathbf{m}^{k}$ that summarizes *all* evidence seen so far. Because $|\mathbf{m}^{k}| = M$ is constant, both compute and memory per step are $O(C + M)$, yielding an overall linear complexity $O(N)$.

Introducing the latent sequence $\mathbf{m}^{1:K-1}$ decomposes the original likelihood as

$$p(\mathbf{x}_{1:N}) = \sum_{\mathbf{m}^{1:K-1}} \prod_{k=1}^{K} \underbrace{p(\mathbf{c}^{k} \mid \mathbf{m}^{k-1})}_{\text{read}}\ \underbrace{p(\mathbf{m}^{k} \mid \mathbf{c}^{k}, \mathbf{m}^{k-1})}_{\text{write}}, \tag{4}$$

with base case $\mathbf{m}^{0} = \varnothing$. Inside each chunk, we still run an ordinary transformer decoder, but conditioned on a *constant* context window $(\mathbf{c}^{k}, \mathbf{m}^{k-1})$. The read path factorizes token-by-token, $p(\mathbf{c}^{k} \mid \mathbf{m}^{k-1}) = \prod_{i=(k-1)C+1}^{kC} p(x_i \mid \mathbf{x}_{1:i-1}, \mathbf{m}^{k-1})$, while the write path generates the next memory in the same autoregressive fashion.

Table 2: Model performance on LongBench-SUM. All values represent recall rates (%). **Bold** marks the highest value and Underline marks the second highest value in each column.

| Model | GOV REPORT | | | | QMSUM | | | |
|---|---|---|---|---|---|---|---|---|
| | ROUGE-1 | ROUGE-2 | ROUGE-L | AVG | ROUGE-1 | ROUGE-2 | ROUGE-L | AVG |
| Qwen2.5-Instruct-32B | 23.67 | 8.46 | 12.57 | 14.90 | 47.77 | 11.29 | 28.17 | 29.08 |
| Qwen2.5-Instruct-14B | 31.19 | 10.96 | 14.96 | 19.04 | 47.53 | 11.46 | 28.28 | 29.09 |
| Qwen2.5-Instruct-7B | 30.91 | 11.68 | 15.20 | 19.26 | 46.64 | 12.01 | 28.33 | 28.99 |
| QwenLong-L1 | 27.60 | 8.20 | 13.07 | 16.29 | 39.44 | 8.24 | 23.56 | 23.74 |
| Qwen2.5-Instruct-14B-1M | 30.58 | 11.93 | 15.51 | 19.34 | 47.31 | 13.13 | 29.07 | 29.84 |
| Qwen2.5-Instruct-7B-1M | 31.02 | 11.47 | 15.30 | 19.26 | 46.72 | 12.33 | 28.66 | 29.24 |
| DS-Distill-Qwen-32B | 26.13 | 8.86 | 12.98 | 15.99 | 39.09 | 8.75 | 23.96 | 23.93 |
| DS-Distill-Qwen-14B | 28.24 | 9.72 | 13.78 | 17.25 | 41.25 | 8.95 | 25.00 | 25.07 |
| DS-Distill-Qwen-7B | 33.30 | 9.39 | 14.59 | 19.10 | 34.33 | 5.97 | 21.57 | 20.62 |
| **RL-MEMAGENT-14B** | **37.16** | 12.03 | **16.23** | **21.80** | **50.21** | 12.70 | **31.27** | **31.39** |
| **RL-MEMAGENT-7B** | 30.28 | **12.37** | 15.37 | 19.34 | 48.49 | **14.41** | 30.91 | 31.27 |

In our formulation, the model's reading and writing operations over the context constitute an Markov Decision Process(MDP) and the objective of RL is to optimize the final reward obtained by this MDP. Therefore, MemAgent's learning objective is to generate a read–write memory trajectory that maximizes the reward, which corresponds to learning an optimal distribution over memory states conditioned on the input context. This further theoretically illustrates the intrinsic unity between our RL formulation and long-text modeling.

# 3 EXPERIMENTS

## 3.1 EXPERIMENTAL SETUP

**Training Details.** To maintain comparability with previous work, we choose Qwen2.5-7B-Instruct and Qwen2.5-14B-Instruct (Yang et al., 2024) as backbone models. We implement the framework for multi-conversation with independent contexts based on verl (Sheng et al., 2024).

We employ a two-stage curriculum RL strategy:

Table 3: Model performance on LongBench-QA.

| Method | 2Wiki | HQA | MuSiQue | NQA | Qasper | AVG |
|---|---|---|---|---|---|---|
| QwenLong-L1-32B | 83.0 | 69.5 | 51.0 | **26.0** | 24.0 | 50.7 |
| Qwen2.5-Instruct-14B-1M | 70.5 | 65.5 | 35.0 | 22.0 | 22.0 | 43.0 |
| Qwen2.5-Instruct-7B-1M | 67.5 | 58.5 | 26.0 | 21.5 | 24.0 | 39.5 |
| DS-Distill-Qwen-32B | 83.5 | 69.0 | 47.5 | 24.0 | 21.0 | 49.0 |
| DS-Distill-Qwen-14B | 83.5 | 67.0 | 42.0 | 22.0 | 22.0 | 47.3 |
| DS-Distill-Qwen-7B | 47.0 | 31.5 | 7.5 | 4.0 | 17.5 | 21.5 |
| Qwen2.5-Instruct-32B | 68.5 | 66.0 | 37.0 | 24.5 | 22.5 | 43.7 |
| Qwen2.5-Instruct-14B | 67.0 | 63.5 | 39.0 | 20.0 | 20.5 | 42.0 |
| Qwen2.5-Instruct-7B | 52.5 | 59.0 | 24.0 | 19.0 | 19.0 | 34.7 |
| MEMAGENT-14B | 79.0 | **73.0** | **52.0** | 25.0 | 26.0 | **51.0** |
| MEMAGENT-7B | 74.0 | 69.5 | 47.0 | 21.5 | **29.0** | 48.2 |

1) stage I focuses on enabling the model to acquire fundamental memory capabilities; 2) stage II trains the model to transfer these capabilities to more diverse contexts and challenging tasks. Specific hyperparameters for training are detailed in § A.3.

During training, we intentionally limit the model to an 8K context window to demonstrate its extrapolation capabilities. This 8K window is allocated as follows: 1024 tokens for the query, 5000 tokens for the context chunk, 1024 tokens for the memory, and 1024 tokens for the output, with the remaining tokens reserved for the chat template.

**Benchmarks.** We conduct comprehensive evaluations on several long-text benchmarks to assess the model's capabilities across various text types and tasks.

1. **RULER-HQA.** This benchmark is created using the same synthetic method as in the first-stage training data. It consists of tasks with a moderate information density and controllable length, where the context distribution is close to natural language, serving as a quantitative evaluation of extrapolation performance.

2. **LongBench-QA.** This benchmark is composed of NarrativeQA (Kočiský et al., 2018), Qasper (Dasigi et al., 2021), HotpotQA (Yang et al., 2018), 2WikiMultihopQA Ho et al. (2020), and MuSiQue (Trivedi et al., 2022). The tasks are relatively short but have a high information density, which severely tests the model's flexible memory management. It also evaluates the model's ability to generalize its memory capabilities to various materials, such as novels, news articles, and Wiki items.

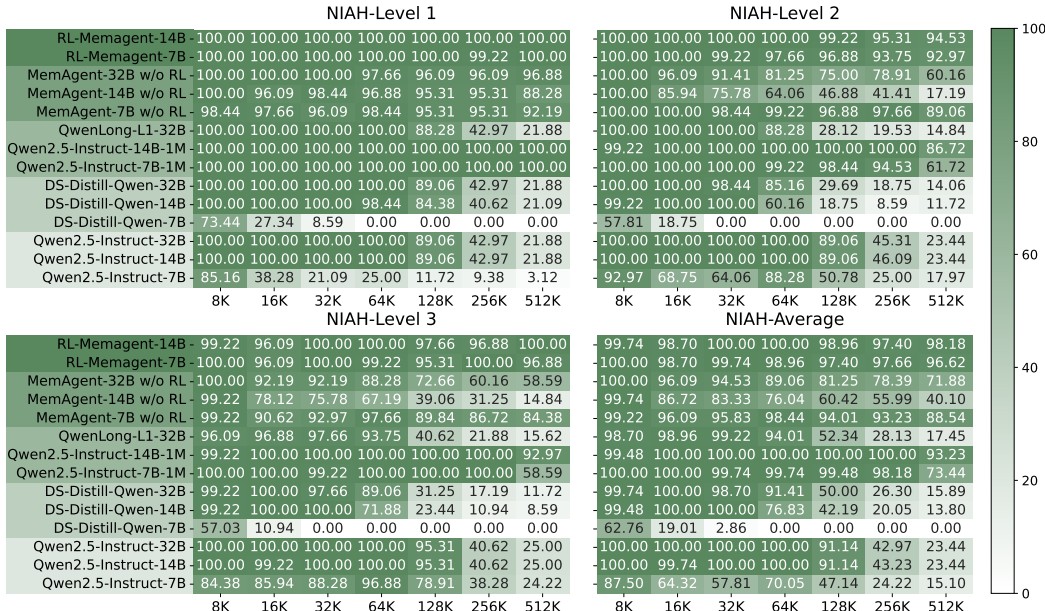

Figure 5: Performance heatmaps on NIAH benchmark across different context lengths.

3. **NIAH.** Needle in a haystack (NIAH) (Kamradt, 2023) is a series of extremely long synthetic tasks with very low information density. To succeed, the model must identify key information and maintain its integrity throughout a long process, thereby testing the robustness of memory.

4. **LongBench-SUM.** We also adopt two long-context summay tasks, GovReport(Huang et al., 2021) and QMSum(Zhong et al., 2021) from LongBench(Bai et al., 2024) to evaluate the performance in different task category that is different from retrieval QA.

**Baselines.** We use DeepSeek-R1-Distill-Qwen (Guo et al., 2025), Qwen-2.5-Instruct-1M (Yang et al., 2025) , Qwen-2.5-Instruct (Yang et al., 2024)and QwenLong-L1 (Wan et al., 2025) as baselines. Their generation configurations are shown in Table 6, while MEMAGENT uses the same context management as described previously in **Training Details**. We also compare MEMAGENT with other agent method, detailed in § D.2.

## 3.2 MAIN RESULTS

**RULER-HQA.** The results are reported in Table 10. We conduct a comparative analysis of all model performances with context lengths ranging from 7K to 896K. For MEMAGENT, we extend the evaluation to ultra-long contexts of 1.75M and 3.5M to assess its extrapolation capabilities.

MEMAGENT exhibits remarkable length extrapolation capabilities with only marginal performance decay as the input context-length increases. In contrast, baseline models show distinct failure patterns. DS-Distill-Qwen series show rapid performance degradation. QwenLong-L1 maintains reasonable performance within its training length but experiences substantial degradation afterward. The Qwen2.5-Instruct-1M series maintains acceptable performance up to 112K tokens, but the performance deteriorates to zero at 896K tokens, well before reaching their theoretical 1M token capacity. This suggests that despite extended context windows, these models struggle with effective information utilization in ultra-long contexts.

**LongBench-QA.** The results on the LongBench-QA benchmark are presented in Table 3. MEMA-GENT demonstrates superior overall performance, outperforming larger long-context or reasoning models. Reasoning models such as the DS-Distill families and the QwenLong model which are trained on a complex dataset, exhibit strong performance. In contrast, the Qwen2.5-Instruct-1M series shows limited improvement over its backbone model. This suggests that LongBench-QA emphasize

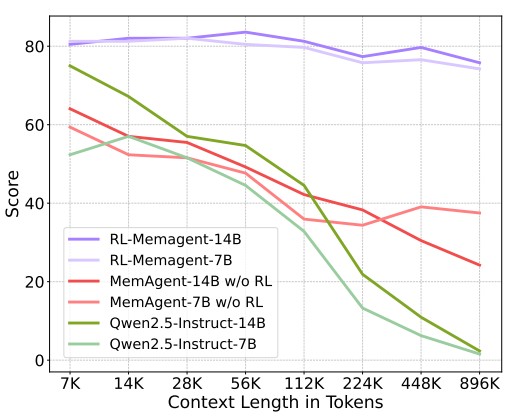 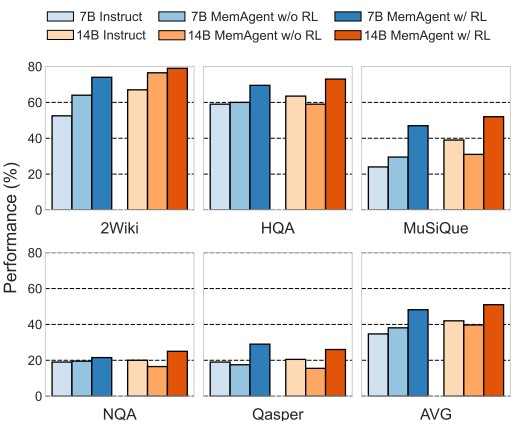

Figure 6: Ablation result of RL training on RULER-HQA.

Figure 7: Ablation result of RL training on Longbench-QA.

a deeper understanding of text rather than simple retrieval ability. The performance of MEMAGENT demonstrates that the memory capabilities acquired through reinforcement learning are generalizable.

**NIAH.** We adopt three variants of NIAH from the RULER benchmark Hsieh et al. (2024) with increasing difficulty across three levels. As depicted in Figure 5, the majority of baselines struggle to maintain consistent performance even within a 128K context window, even Qwen2.5-Instruct-1M also experience a performance drop at 512K. RL-MEMAGENT, despite suffering some performance fluctuations, shows only a minimal performance loss of less than 5% at 512K. This robust performance is particularly noteworthy given that the evaluation at 512K involves more than 100 turns of dialogue.

**LongBench-SUM.** We evaluate summary quality by the recall scores of `ROUGE-{1,2,L}`. RL-MEMAGENT achieves SOTA on almost all metrics, demonstrating that the model has learned general memory and context management capabilities, rather than abilities specific to the QA task.

## 3.3 ABLATION STUDY

### 3.3.1 RL TRAINING

To investigate the impact of reinforcement learning, we conduct ablation experiments. The results of RULER-HQA and NIAH are presented in Figure 6 and Figure 5, respectively. MEMAGENT without reinforcement learning training outperforms the backbone models; however, it still exhibits a substantial decline in performance as the input length increases. The results of Longbench-QA, shown in Figure 7, demonstrate that directly applying MEMAGENT leads to only marginal or even negative improvements. In contrast, RL-MEMAGENT achieves significant improvements in both evaluation scenarios, indicating reinforcement learning training is essential to develop generalizable and robust memory abilities.

### 3.3.2 MEMORY LENGTH

Selecting an appropriate MEMAGENT setting involves certain trade-offs. A larger memory size allows the model to store more useful information, but it also introduces challenges in memory management and increases the likelihood of redundancy. Conversely situation may lead to insufficient storage capacity, leaving the model without the necessary references.

To achieve a reasonable compression ratio while keeping the total context length within 8,192 tokens, we set the default configuration of MEMAGENT to use a 1,024-token memory and context chunks of 5,000 tokens, based on preliminary validation results.

To investigate the effect of hyperparameter choices, we conduct an ablation study on memory length ranging from 256 to 4096. The results presented in Figure 8 and Figure 9, showing that our chosen configuration constitutes a reasonable sweet spot, and that MEMAGENT 's performance is robust over different memory size. We further examine the impact of varying the context size in § D.1 and observe similar trends.

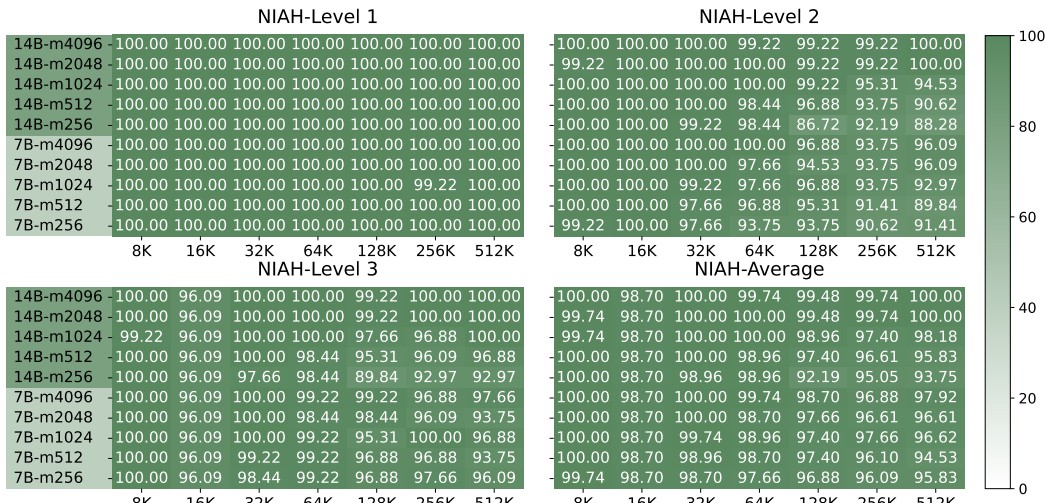

Figure 8: Ablation result of memory-length on NIAH

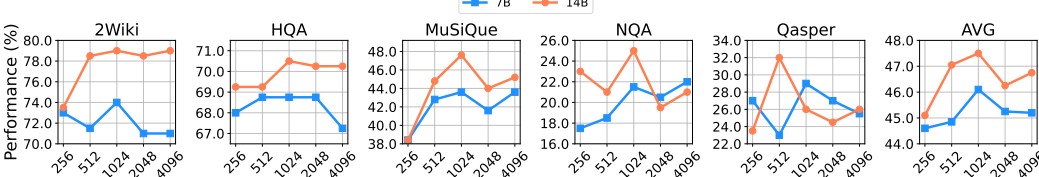

Figure 9: Ablation result of memory-length on Longbench

### 3.3.3 CONTEXT DISTRIBUTION

Although our experiments show that MEMAGENT can effectively extrapolate to a length of 3.5M tokens, we still wish to examine whether MEMAGENT is affected by issues such as information-overwritten and the lost-in-the-middle phenomenon. Our hypothesis is that overcoming such problem is a natural result of end-to-end optimization. During training, the model learns to preserve and track critical information in order to maximize the final reward.

To validate this hypothesis, we carefully design a set of probing experiments based on RULER-HQA, where the context is consist of some key information and many distractors. We divided the key information into two groups and placed them at different positions within the context. We constructed five settings: (0%, 100%), (20%, 80%), (40%, 60%), (0%, 20%), and (80%, 100%), where 0% indicates the beginning of the context and 100% means the end of the context.

For example, in the (0%, 100%) case, the model sees one piece of key information at the very beginning and the other only at the final memory update step. This represents one of the most challenging scenarios for the information-overwritten problem. While (40%, 60%) may serve as a challenging lost-in-the-middle setting.

The results shown in table 4 indicates that MemAgent remains consistently robust across all patterns without exhibiting any catastrophic performance degradation. This strongly supports our hypothesis that the general memory abilities acquired through trial and error are not tied to any particular context pattern.

## 4 RELATED WORK

**Long Context LLMs.** Extrapolation methods for RoPE-based LLMs (Su et al., 2024), such as NTK (bloc97, 2023), PI (Chen et al., 2023), YaRN (Peng et al., 2023b) and DCA (An et al., 2024),

Table 4: Probe experiment results. **Ctx. Dist.** denotes the context distribution, where the two numbers correspond to the relative positions of the two key-information groups within the entire context. 0% means the beginning and 100% means the end. **random** indicates randomly shuffling all context items, consistent with the setup in the main experiment. The other rows show the performance difference relative to **random**. All values represent accuracy (%).

| Model | Ctx. Dist. | Length | | | | | | | |
|---|---|---|---|---|---|---|---|---|---|
| | | 7K | 14K | 28K | 56K | 112K | 224K | 448K | AVG |
| 14B | random | 80.47 | 82.03 | 82.03 | 83.59 | 81.25 | 77.34 | 79.69 | 75.78 |
| | 0% 20% | +3.91 | -3.91 | +3.13 | +1.57 | 0.00 | +4.69 | +3.90 | +1.90 |
| | 0% 100% | +3.12 | +0.78 | +3.13 | -3.12 | +1.56 | +7.82 | +6.25 | +2.79 |
| | 20% 80% | +0.78 | -3.12 | +2.35 | +0.79 | -3.13 | +5.47 | -3.13 | 0.00 |
| | 40% 60% | +1.56 | +2.35 | -2.34 | -3.12 | -1.56 | +3.13 | -0.78 | -0.11 |
| | 80% 100% | -2.35 | 0.00 | +1.56 | +0.79 | +3.13 | 0.00 | +1.56 | +0.67 |
| 7B | random | 81.25 | 81.25 | 82.03 | 80.47 | 79.69 | 75.78 | 76.56 | 79.58 |
| | 0% 20% | -1.56 | -0.78 | +3.13 | +3.91 | +3.12 | +5.47 | +3.13 | +2.35 |
| | 0% 100% | +0.78 | 0.00 | +2.35 | +1.56 | +3.90 | +4.69 | +2.35 | +2.23 |
| | 20% 80% | -0.78 | -0.78 | +2.35 | 0.00 | 0.00 | 0.00 | +3.13 | +0.56 |
| | 40% 60% | 0.00 | +1.56 | +3.13 | +0.78 | -3.91 | +3.13 | +5.47 | +1.45 |
| | 80% 100% | +1.56 | 0.00 | +0.78 | -0.78 | +0.78 | +4.69 | 0.00 | +1.00 |

modify the components of positional embeddings, enabling the model to capture long-range semantic dependencies. On the other hand, Linear attention mechanisms (Child et al., 2019; Katharopoulos et al., 2020), Recurrent Neural Networks (RNNs) and State Space Models (SSMs) (Gu et al., 2021; Gu & Dao, 2023; Peng et al., 2023a; De et al., 2024; Feng et al., 2024), sparse attention (Beltagy et al., 2020; Zhao et al., 2019; Xiao et al., 2023; Yuan et al., 2025; Lu et al., 2025) focus on architecture improvements. Chunk strategy have also been explored in long-context modeling (Li et al., 2025b; Liao et al., 2025), while MEMAGENT aims to equip memory ability to any backbone model via post-training with standard RL frameworks without heavily changing on architecture.

**Memory Mechanism.** The Long Short-Term Memory (LSTM) mechanism (Hochreiter & Schmidhuber, 1997) achieved significant success in early NLP tasks, while Neural Turing Machines (Graves et al., 2014) and Memory Networks (Weston et al., 2014) demonstrated how to equip neural networks with memory. Existing memory mechanisms integrated to Transformer models are typically realized by adding external memory modules (Martins et al., 2021; Wu et al., 2020; Behrouz et al., 2024; Bulatov et al., 2023) or external database (Zhong et al., 2024; Lu et al., 2023; Modarressi et al., 2023). Recently, retrieval-augmented memory agent (Fang et al., 2025; Chhikara et al., 2025; Zhou et al., 2025) workflows have attracted the community's attention. The diffrence between MEMAGENT and other agent is that we use reinforcement learning to enable LLM itself the ability to memorize.

**Reinforcement Learning for LLMs.** In recent RL studies, the reward signals have gradually shifted from human preferences (Ouyang et al., 2022) or reward models distilled from them (Bai et al., 2022) to rule-based feedback, which has demonstrated great potential in enhancing model reasoning capabilities (OpenAI, 2024; Guo et al., 2025; Qwen, 2024; DeepMind, 2024; Team et al., 2025) with GAE (Schulman et al., 2018) based PPO (Schulman et al., 2017) or GRPO (Shao et al., 2024) training. Algorithmic enhancements (Hu, 2025; Yu et al., 2025; Liu et al., 2025) have mostly focused on improving sustainability and efficiency of these algorithms. To further release the potential of RL, recent works such as Search-R1 (Jin et al., 2025), Agent-R1 (Ouyang et al., 2025) and RAGEN (Wang et al., 2025) have explored the training of tool-using agents based on multi-turn chat. GiGPO (Feng et al., 2025) further investigates the use of multiple independent contexts in agent training.

## 5 CONCLUSION

In this paper, we introduce MEMAGENT, a novel long-context method that employs an RL-trained memory module, which enables large language models (LLMs) to selectively record relevant information while disregarding extraneous details. Our experiments demonstrate that when trained on 60K-length sequences, MEMAGENT exhibits remarkable extrapolation, extending its effective context to 3.5M tokens with only 8K context. The model achieves state-of-the-art performance across a diverse range of long-context tasks. Our ablation studies reveal the critical role of RL-based training in achieving these results and how memory capacity influences performance across different task types, providing key insights into the proposed memory mechanism. We hope that this work may lay a strong foundation for developing more advanced memory architectures and training strategies, thereby paving the way for significantly enhancing the long-context capabilities of LLMs.

## REPRODUCIBILITY STATEMENT

For reproducibility, we have provided the inplementation details in (§ A), including the prompt template (§ A.1), pseudocode (§ A.2) and training recipe and algorithm hyperparameter (§ A.3) and evaluation settings (§ A.4). The training and evaluation code, as well as the dataset and model weights, will be available in open-source platforms.

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

# A    IMPLEMENTATION DETAILS

## A.1    PROMPT TEMPLATE

---

You are presented with a **problem**, a **section** of an article that may contain the answer, and a **previous memory**. Please read the section carefully and *update the memory* with new information that helps to answer the problem, while retaining all relevant details from the previous memory.

```
<problem> {prompt} </problem>
<memory> {memory} </memory>
<section> {chunk} </section>
```

**Updated memory:**

---

You are presented with a **problem** and a **previous memory**. Please answer the problem based on the previous memory and put the answer in \boxed {}.

```
<problem> {prompt} </problem>
<memory> {memory} </memory>
```

**Your answer:**

---

Table 5: Template of MEMAGENT for context processing (top part) and final answer generation (bottom). Curly-brace placeholders {} will be replaced with actual content.

## A.2    ALGORITHM

---

**Algorithm 1** Multi-conv DAPO

---

**Require:** Policy model $\pi_\theta$, reference model $\pi_{\text{ref}}$ (frozen), rule-based verifier $V$, dataset $\mathcal{D}$, group size $G$

1: **while** not converged **do**
2:     Sample a prompt $q \sim \mathcal{D}$
3:     **for** $g = 1$ to $G$ **do**                                  ▷ Group of conversations for the same $q$
4:         Initialize $h_{g,0} \leftarrow [q]$
5:         **for** $t = 1$ to $c_g$ **do**                          ▷ Context-independent conversation
6:             Sample $o_{g,t} \sim \pi_\theta(\cdot \mid h_{g,t-1})$
7:             $h_{g,t} \leftarrow h_{g,t-1} \parallel o_{g,t}$
8:         **end for**
9:         $y_g \leftarrow o_{g,c_g}$                                ▷ Final response used for scoring
10:        $\hat{r}_g \leftarrow V(q, y_g)$                          ▷ Rule-based reward
11:        $d_g \leftarrow \text{KL}\big(\pi_\theta(\cdot \mid h_{g,c_g}) \parallel \pi_{\text{ref}}(\cdot \mid h_{g,c_g})\big)$
12:        $r_g \leftarrow \hat{r}_g - \beta d_g$
13:    **end for**
14:    $\{A_g\}_{g=1}^G \leftarrow \text{GroupNorm}(\{r_g\}_{g=1}^G)$
15:    **for** $g = 1$ to $G$ **do**
16:        $\rho_g \leftarrow \dfrac{\pi_\theta(y_g \mid h_{g,c_g})}{\pi_{\theta_{\text{old}}}(y_g \mid h_{g,c_g})}$
17:        $\mathcal{J}_g \leftarrow \min\big(\rho_g A_g, \text{clip}(\rho_g, 1 - \epsilon_{low}, 1 + \epsilon_{high}) A_g\big)$
18:    **end for**
19:    $\mathcal{J}_{\text{clip}} \leftarrow \frac{1}{G} \sum_{g=1}^G \mathcal{J}_g$
20:    $\theta \leftarrow \theta + \eta \nabla_\theta \mathcal{J}_{\text{clip}}$
21: **end while**

---

## A.3    TRAINING

We use the DAPO algorithm for training, applying a KL factor of $1 \times 10^{-3}$ and disabling the entropy loss. The AdamW optimizer is employed with a constant learning rate of $1 \times 10^{-6}$ and a linear warm-up scheduler, with the wram-up step set to 20. We use a rollout batchsize of 256, with a group size of 16. Note that due to the multi-conversation feature of MEMAGENT, the actual `mini-batchsize`

is not equal to `rollout batchsize`/16. We utilize off-policy training by fixing the ratio of the sample batch size to the backpropagation batch size is set to 16.

We shift to stage II when stage I are fully converged, which takes about 400 steps. Here is the training data recipe of each stage.

- **Stage I** We use 32,768 synthetic QA data instances, each approximately 32K tokens in length. These are based on the HotpotQA (Yang et al., 2018) dataset and follow the RULER (Hsieh et al., 2024) methodology, which involves embedding golden paragraphs (containing correct answers) within extensive distractor content sampled from the same dataset.

- **Stage II** We use 2,560 training instances with a maximum length of 60K tokens. This set consists of difficult, high-quality long-text QA data from DocQA-RL-1.6K (Wan et al., 2025), mixed with data from the first stage.

Each training sample used in stage I is of 200 articles in HotpotQA, with an approximate total token length of 28K. We thoroughly clean the dataset by filtering out questions where Qwen2.5-7B-Base or Qwen2.5-7B-Instruct achieves 100 % Best-Of-2 score **without given any context**. These questions likely represent common knowledge already internalized within the models' memories. 80,000 samples from the HotpotQA training split are processed through this pipeline and approximately 50% of the data are filtered out. We chose the frist 32,768 samples of processed data as our training set.

We then apply a similar approach to synthesize 128 samples from the HotpotQA validation set. For extrapolation performance testing, we synthesize test sets with different context lengths using the same pipeline. The number of wiki items ranges from 50 up to 6400, corresponding to context lengths of approximately 7K to 3.5M tokens.

## A.4 EVALUATION

We extract answers from the model outputs using regular expressions, and we prompt the model to respond in the specified format. The chosen format is 'the answer is ANSWER.'

We employ the `sub_em` score for all benchmarks. This means that an answer is considered correct if it contains all the elements of the ground truth. When an answer consists of multiple parts and the expected response should include all of them, the score corresponds to the proportion of correct parts provided.

Before evaluating the answers, we normalize both the ground truth and the extracted responses. For example, we remove definite articles, ignore case distinctions, and apply similar standard normalization steps following previous work (Wan et al., 2025; Hsieh et al., 2024; Yen et al., 2024).

Table 6 shows the generation configurations of baseline models.

Table 6: Generation configurations of baseline models.

| Model | Context Length | Input/Output Tokens |
|---|---|---|
| QwenLong-L1 (Wan et al., 2025) | 128K | 120,000 / 10,000 |
| Qwen2.5-Instruct-1M Series (Yang et al., 2025) | 1M | 990,000 / 10,000 |
| DeepSeek-R1-Distill-Qwen Series (Guo et al., 2025) | 128K | 120,000 / 10,000 |
| Qwen2.5-Instruct Series(Yang et al., 2024) | 128K | 120,000 / 10,000 |

**NIAH** `niah_single_{1,2,3}` in RULER (Hsieh et al., 2024) benchmark are used in our test. The yaml configuration of RULER are presented in 7. In level 1, the "haystack" consists of repetitive sentences, and the "needle" is a seven-digit number associated with a magic word. For level 2, the "haystack" is composed of longer essays. Level 3 goes a step further than Level 2 where the "needle" is a 36-character UUID string. Question and context are concated as the input of LLMs. We omit the `answer_prefix` provided in original RULER benchmark since it is not compatible with MEMAGENT workflow.

```
niah_single_1:
  task:  niah
  args:
    type_haystack:  repeat
    type_needle_k:  words
    type_needle_v:  numbers
    num_needle_k:  1
    num_needle_v:  1
    num_needle_q:  1
niah_single_2:
  task:  niah
  args:
    type_haystack:  essay
    type_needle_k:  words
    type_needle_v:  numbers
    num_needle_k:  1
    num_needle_v:  1
    num_needle_q:  1
niah_single_3:
  task:  niah
  args:
    type_haystack:  essay
    type_needle_k:  words
    type_needle_v:  uuids
    num_needle_k:  1
    num_needle_v:  1
    num_needle_q:  1
```

Table 7: Synthetic Configuration used for NIAH task.

# B    COMPUTATION COMPLEXITY

We adopt the floating-point operations (FLOP) estimator for the Qwen2Model from verl Sheng et al. (2024) to compute the FLOP cost of both the baseline model and our proposed method. The results are shown in Figure 10. The baseline model exhibits an $O(n^2)$ complexity, while MEMAGENT achieves an $O(n)$ complexity.

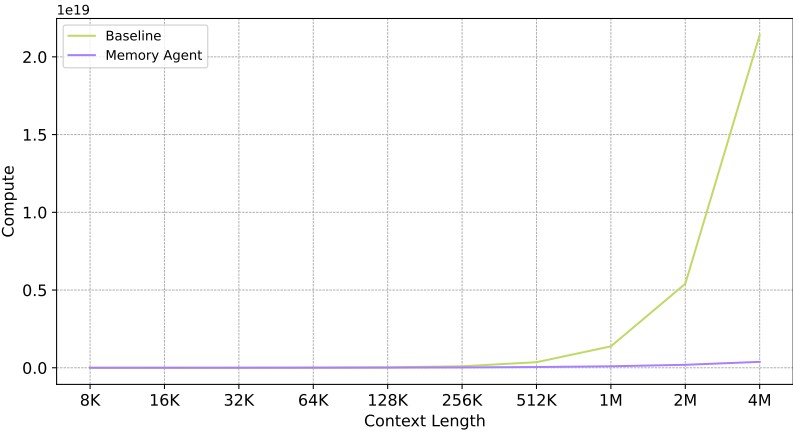

Figure 10: Floating point operations across context lengths from 8K to 4M

For the baseline model, the number of tokens required to process is $q + c + o$, where $q$ represents the length for the problem, $c$ is the context length and $o$ represents the output length.

For MEMAGENT, total FLOP cost is the sum of the FLOPs from all stages. The detailed stages involved are as follows:

- Initializing: In the first stage, the model processes an input consisting of $q + 200 + o$, where 200 represents a constant added to prompt the model to follow the MEMAGENT workflow.

- Memory Updating: The number of repetitions is determined by $k = \lceil \frac{c}{N} \rceil$, where $c$ is the variable component of the input. Each repetition requires an input of length $q + 200 + N + o$.

- Final Answering: The final stage processes an input of length $q + 100 + o$, which includes the accumulated output from the previous steps.

We set $q = 1024, o = 1024, N = 5000$ and $c$ is ranging from 8K to 4M to calculate the final result.

## C  CASE STUDY

To further illustrate the proposed memory mechanism in detail, we conduct a case study on a generation trajectory of MEMAGENT-14B. The input question is: *The director of the romantic comedy 'Big Stone Gap' is based in what New York city?* This a 2-hop question with the following relevant Wikipedia entries:

1) **Big Stone Gap** is a 2014 American drama romantic comedy film written and directed by Adriana Trigiani.

2) **Adriana Trigiani** is an Italian American best-selling author of sixteen books, television writer, film director, and entrepreneur based in Greenwich Village, New York City.

In the first round, the model is presented with the entry *Ghost*, which refers to a production team also based in *New York*. The model chooses to retain this potentially useful information for future use. In the second round, no relevant context is provided; nevertheless, the model maintains its agent state, demonstrating robustness against distraction. In the third round, both relevant entries are presented. The model correctly identifies critical information and updates its memory accordingly, leading to the correct answer: *Greenwich Village, New York City*. At this point, the reasoning process is complete. In the remaining rounds, the model's memory remains unchanged and is used to produce the final response.

---

CHUNK1

...
Document 20: Ghost (production team)
Ghost is a Swedish record producing and songwriting team, composed of Ulf Lindström and Johan Ekhé, based in *New York City*.
...

MEMORY1
The provided section does not contain any information about the director of the romantic comedy "Big Stone Gap" or their base in New York City. **However, it does mention a production team called "Ghost" which is based in New York City.** Since the problem is specifically asking about the director of "Big Stone Gap," and there is no direct information about the director in the given section, we cannot infer anything directly related to the problem from the provided text.
Updated memory:

- Ghost is a Swedish record producing and songwriting team, composed of Ulf Lindström and Johan Ekhé, based in New York City. They are perhaps best known for writing and producing Swedish singer Robyn's three first studio albums.

The problem remains unanswered based on the provided section. Additional information would be needed to determine the director of "Big Stone Gap" and their base in New York City.

---

---

CHUNK2
...(No relevant document)

MEMORY2
**The provided section does not contain any information about the director of the romantic comedy "Big Stone Gap" or their base in New York City. 2 Therefore, the updated memory remains unchanged with respect to the problem at hand.** The memory still notes that the Ghost record producing and songwriting team is based in New York City, but it does not provide any information about the director of "Big Stone Gap."
Updated memory:

- Ghost is a Swedish record producing and songwriting team, composed of Ulf Lindström and Johan Ekhé, based in New York City. They are perhaps best known for writing and producing Swedish singer Robyn's three first studio albums.

- The problem remains unanswered based on the provided section. Additional information would be needed to determine the director of "Big Stone Gap" and their base in New York City.

---

CHUNK3
...
Document 81: Big Stone Gap (film)
**Big Stone Gap** is a 2014 American drama romantic comedy film written and directed by **Adriana Trigiani** and produced by Donna Gigliotti for Altar Identity Studios, a subsidiary of Media Society.
...
Document 96: Adriana Trigiani
**Adriana Trigiani** is an Italian American best-selling author of sixteen books, television writer, film director, and entrepreneur based in **Greenwich Village, New York City**.
...

MEMORY3
Updated memory:

- Ghost is a Swedish record producing and songwriting team, composed of Ulf Lindström and Johan Ekhé, based in New York City. They are perhaps best known for writing and producing Swedish singer Robyn's three first studio albums.

- **Adriana Trigiani** is an Italian American best-selling author of sixteen books, television writer, film director, and entrepreneur based in Greenwich Village, New York City. Trigiani has published a novel a year since 2000.

- **Big Stone Gap** is a 2014 American drama romantic comedy film written and directed by Adriana Trigiani and produced by Donna Gigliotti for Altar Identity Studios, a subsidiary of Media Society. Based on Trigiani's 2000 best-selling novel of the same name, the story is set in the actual Virginia town of Big Stone Gap circa 1970s. The film had its world premiere at the Virginia Film Festival on November 6, 2014.

  The director of the romantic comedy "Big Stone Gap" is Adriana Trigiani, who is based in **Greenwich Village, New York City**.

---

This trajectory highlights several core capabilities learned by the memory agent: estimating potentially relevant content from query keywords and storing it preemptively, immediately updating memory upon encountering context that matches the query, and remaining unaffected by irrelevant information. Notably, these memory behaviors are not the result of architectural attention mechanisms, but emerge as text generation abilities reinforced through RL.

# D SUPPLEMENTARY EXPERIMENTS

## D.1 ABLATION STUDY ON CONTEXT SIZE AND MEMORY SIZE

In this ablation experiments, we keep the total length of memory size + context chunk size constant, and linearly adjusted the memory size. The goal is to control the total context length per conversation turn. The results are shown in Table 8 and Table 9.

Table 8: Ablation result of memory-size and context-size on NIAH. All number indicates the averaging score of level1̃level3

| Method | 8K | 16K | 32K | 64K | 128K | 256K | 512K |
|---|---|---|---|---|---|---|---|
| 14B-m4096-c1928 | 100.00 | 99.22 | 98.70 | 97.66 | 98.96 | 99.22 | 97.92 |
| 14B-m3072-c2952 | 99.74 | 100.00 | 100.00 | 100.00 | 98.70 | 99.48 | 99.48 |
| 14B-m2048-c3976 | 99.74 | 99.48 | 99.48 | 100.00 | 100.00 | 97.13 | 97.66 |
| 14B-m1024-c5000 | 99.74 | 98.70 | 100.00 | 100.00 | 98.96 | 97.40 | 98.18 |
| 7B-m4096-c1928 | 100.00 | 99.22 | 98.44 | 96.35 | 98.96 | 98.18 | 97.14 |
| 7B-m3072-c2952 | 99.48 | 100.00 | 99.74 | 99.48 | 96.88 | 98.18 | 96.09 |
| 7B-m2048-c3976 | 99.74 | 99.48 | 99.48 | 99.48 | 98.96 | 94.53 | 94.53 |
| 7B-m1024-c5000 | 100.00 | 98.70 | 99.74 | 98.96 | 97.40 | 97.66 | 96.62 |

Table 9: Ablation result of memory-size and context-size on Longbench-QA

| Method | 2Wiki | HQA | MuSiQue | NQA | Qasper | AVG |
|---|---|---|---|---|---|---|
| 14B-m4096-c1928 | 74.5 | 72.5 | 48.5 | 21.5 | 25.5 | 48.5 |
| 14B-m3072-c2952 | 76.5 | 70.5 | 52.5 | 24.5 | 26.5 | 50.1 |
| 14B-m2048-c3976 | 74.5 | 71.5 | 49.5 | 23.0 | 27.0 | 49.1 |
| 14B-m1024-c5000 | 79.0 | 73.0 | 52.0 | 25.0 | 26.0 | 51.0 |
| 7B-m4096-c1928 | 70.0 | 66.0 | 45.5 | 19.0 | 26.0 | 45.3 |
| 7B-m3072-c2952 | 72.0 | 64.0 | 44.0 | 20.0 | 25.5 | 45.1 |
| 7B-m2048-c3976 | 75.0 | 69.0 | 43.5 | 23.0 | 26.0 | 47.3 |
| 7B-m1024-c5000 | 74.0 | 69.5 | 47.0 | 21.5 | 29.0 | 48.2 |

## D.2 AGENT BASELINES

We compare MemAgent against an advanced memory-agent method, Mem0(Chhikara et al., 2025). The Mem0 paper also reports that RAG methods using only top-1 or top-2 retrieval form strong and stable baselines for memory-agent tasks. Therefore, we conduct extensive comparisons against RAG agents under multiple configurations.

For Mem0, we use SOTA OpenAI models, GPT-5.1 and text-embedding-3-large as langugae model and embedding model respectively and we follow the official GitHub repository for memory updating and retrieval. Specifically, during memory creation, we split and processed the entire context in 5,000-token chunks; during retrieval, we selected the top 30 memories.

For RAG, we also use text-embedding-3-large as embedding model and configure it with various chunk size and top-K value.

The results show that MEMAGENT outperforms these methods, demonstrating that end-to-end RL–trained memory provides greater flexibility and coherence compared with retrieval-based strategies.

Table 10: Result versus RAG Agent in RULER-HQA with different top-**K** settings. We segment the context based on natural semantic units, i.e., each wiki item was treated as a chunk.

| Model | Length | | | | | | | | | |
|---|---|---|---|---|---|---|---|---|---|---|
| | 7K | 14K | 28K | 56K | 112K | 224K | 448K | 896K | 1.75M | 3.5M |
| RAG + Qwen2.5-14B | | | | | | | | | | |
| *K=2* | 57.03 | 54.69 | 51.56 | 54.69 | 53.12 | 50.00 | 52.34 | 49.22 | 48.44 | 48.44 |
| *K=4* | 66.41 | 67.19 | 68.75 | 67.19 | 66.41 | 64.06 | 66.41 | 64.84 | 60.94 | 59.38 |
| *K=6* | 72.66 | 75.78 | 75.78 | 74.22 | 69.53 | 71.88 | 73.44 | 67.19 | 65.62 | 66.41 |
| *K=8* | 78.12 | 78.91 | 77.34 | 81.25 | 76.56 | 78.12 | 77.34 | 74.22 | 70.31 | 64.84 |
| **RL-MEMAGENT-14B** | **80.47** | **82.03** | **82.03** | **83.59** | **81.25** | **77.34** | **79.69** | **75.78** | **78.91** | **71.09** |
| RAG + Qwen2.5-7B | | | | | | | | | | |
| *K=2* | 53.91 | 54.69 | 53.12 | 51.56 | 54.69 | 51.56 | 52.34 | 49.22 | 48.44 | 46.09 |
| *K=4* | 67.19 | 66.41 | 66.41 | 67.19 | 64.84 | 64.06 | 62.50 | 61.72 | 60.94 | 59.38 |
| *K=6* | 74.22 | 73.44 | 72.66 | 73.44 | 70.31 | 73.44 | 70.31 | 67.19 | 65.62 | 65.62 |
| *K=8* | 75.00 | 75.00 | 75.78 | 74.22 | 74.22 | 77.34 | 72.66 | 68.75 | 64.06 | 64.84 |
| **RL-MEMAGENT-7B** | **81.25** | **81.25** | **82.03** | **80.47** | **79.69** | **75.78** | **76.56** | **74.22** | **77.34** | **71.88** |

Table 11: Result versus RAG Agent in Longbench-QA with different top-**K** and **C**ontext size settings. We segment the context using fixed-length chunks. For retrieval, we performed top-k matching using cosine similarity scores.

| Method | 2Wiki | HQA | MuSiQue | NQA | Qasper | AVG |
|---|---|---|---|---|---|---|
| Qwen2.5-14B + RAG | | | | | | |
| *C=1024 K=2* | 51.50 | 56.50 | 26.50 | 15.00 | 23.50 | 28.83 |
| *C=1024 K=4* | 70.00 | 64.50 | 34.50 | 17.50 | 27.00 | 35.58 |
| *C=1024 K=6* | 71.50 | 64.00 | 41.00 | 19.00 | **27.50** | 37.17 |
| *C=1024 K=8* | 72.50 | 64.50 | 39.00 | 17.50 | 26.00 | 36.58 |
| *C=2048 K=2* | 58.50 | 61.50 | 33.50 | 13.50 | 25.50 | 32.08 |
| *C=2048 K=4* | 76.00 | 64.00 | 36.00 | 18.50 | 25.00 | 36.58 |
| *C=2048 K=6* | 73.00 | 67.50 | 41.50 | 21.00 | 26.50 | 38.25 |
| *C=2048 K=8* | 77.50 | 68.50 | 42.00 | 21.00 | **27.50** | 39.42 |
| **RL-MemAgent-14B** | **79.0** | **73.0** | **52.00** | **25.00** | 26.00 | **51.00** |
| Qwen2.5-7B + RAG | | | | | | |
| *C=1024 K=2* | 41.00 | 48.50 | 22.00 | 14.50 | 25.50 | 25.25 |
| *C=1024 K=4* | 49.00 | 56.50 | 28.00 | 17.00 | 28.50 | 29.83 |
| *C=1024 K=6* | 54.50 | 57.50 | 29.50 | 17.00 | 25.00 | 30.58 |
| *C=1024 K=8* | 50.50 | 59.00 | 29.50 | 18.00 | 25.00 | 30.33 |
| *C=2048 K=2* | 49.50 | 51.50 | 19.50 | 12.50 | 27.00 | 26.67 |
| *C=2048 K=4* | 50.50 | 53.00 | 26.00 | 17.00 | 25.50 | 28.67 |
| *C=2048 K=6* | 50.50 | 56.50 | 27.50 | **22.00** | 25.50 | 30.33 |
| *C=2048 K=8* | 50.50 | 58.00 | 25.50 | 21.00 | 27.00 | 30.33 |
| **RL-MemAgent-7B** | **74.00** | **69.50** | **47.00** | 21.50 | **29.00** | **48.20** |

# E    LLM USAGE

In this section, we report the usage of LLMs in this work. Some sentences in this manuscript are drafted or refined by LLMs, but all text is finalized by human authors. In the experimental process, LLMs assist with code completion, but they do not produce novel ideas or complete experiments.

# F  FALIURE PATTERN STUDY

## F.1  INFORMATION OVERWRITTEN

In this example, the model accumulates a large amount of irrelevant memories in the early stage (Turn 60). When the crucial context appears, the model does capture the relevant information (Turn 289), but attempts to append it to the end (Turn 290), where it is truncated due to insufficient memory. Later, the model proactively performs a summarization (Turn 317), which gives it more space to store the second key piece of context (Turn 433). However, since the information from the first context has already been lost, the model incorrectly relies on irrelevant information for reasoning and fails to answer the question correctly.

---

**Question:** Where is the company that Sachin Warrier worked for as a software engineer headquartered?

**Relevant Information:**

(shown in Turn 289) Tata Consultancy Services Limited (TCS): An Indian multinational information technology (IT) service, consulting and business solutions company. It is a subsidiary of the Tata Group, operates in 46 countries, and is **headquartered in Mumbai, Maharashtra**.

(shown in Turn 433) Sachin Warrier: A playback singer and composer in the Malayalam cinema industry from Kerala. He became notable with the song "Muthuchippi Poloru" from the film Thattathin Marayathu, made his debut with Malarvaadi Arts Club, and was once working as a software engineer at Tata Consultancy Services in Kochi (later resigned to focus on music). His latest work is as a composer for the movie Aanandam.

**Answer:** Mumbai

---

**Memory Diff between Turn 59 and Turn 60**

-The provided section does not contain any information about Sachin Warrier or his work as a software engineer. Therefore, the updated memory remains unchanged with respect to Sachin Warrier. However, it does provide information about Texas Monthly, which is headquartered in Downtown Austin, Texas. Since the problem is specifically asking about a company where Sachin Warrier worked as a software engineer, and no information about Sachin Warrier is provided, the updated memory remains as follows:
+The provided section does not contain any information about Sachin Warrier or his work as a software engineer. However, it does provide information about Procter & Gamble, which is an American consumer goods corporation headquartered in downtown Cincinnati, Ohio, United States of America. Since the problem is specifically asking about a company where Sachin Warrier worked as a software engineer, and no information about Sachin Warrier is provided, the updated memory remains as follows:

- Pizza Pizza Ltd. is a franchised Canadian pizza fast-food restaurant, with its headquarters in Etobicoke, Toronto. Its restaurants are mainly in the province of Ontario. Other locations operate in Quebec, Nova Scotia, and in western Canada. Franchises in western Canada are mostly run through Alberta-based subsidiary Pizza 73, and in non-traditional locations such as university campuses and movie theatres throughout Canada. It has over 500 locations, including over 150 non-traditional locations.

- TI Automotive develops, manufactures, and supplies automotive fluid storage, carrying, and delivery systems. The company has 23,000 employees at 120 locations in 29 countries, and supplies all of the world's major automobile manufacturers. TI Automotive serves the automotive aftermarket through Bundy, Walbro, and Marwal brands. The company's headquarters are located in Oxford, UK, with Corporate Offices based in Auburn Hills, Michigan. In 2015 TI Automotive was acquired by Bain Capital.

  (...)

- -

**Memory Diff between Turn 282 and Turn 289**

-The provided section does not contain any information about Sachin Warrier or the company he worked for as a software engineer. Therefore, the updated memory remains as follows:

+The provided section contains information about Tata Consultancy Services Limited (TCS), which is an Indian multinational information technology (IT) service, consulting and business solutions company. It states that TCS is headquartered in **Mumbai, Maharashtra.**

+Updated memory:

- Cosmos (formerly Cosmos Tours) is a UK independent tour operator providing a range of package holidays to the UK market. The tour operator is connected to the international Globus Travel Group, founded in 1928, which remains family owned with headquarters in Lugano, Switzerland. Under parent company Cosmos Tours Ltd, the tour operator offers holidays through three brands: Cosmos, Archers Holidays and Avalon Waterways.
- L&F Jones Holdings Ltd is a company based in Westfield, Somerset, England, that includes a convenience store chain, a wholesale food business, and a Best Western PLUS hotel. The store chain comprises twelve shops in Somerset, Wiltshire, Dorset, South Gloucestershire, and Bristol.

    (...)

- Round Table Pizza is a large chain of pizza parlors in the western United States. The company's headquarters are located in Concord, California.

- Round Table Pizza is a large chain of pizza parlors in the western United States. The company's

- Anton Media Group, formerly Anton Community

---

**Memory Diff between Turn 289 and Turn 290**

-The provided section contains information about Tata Consultancy Services Limited (TCS), which is an Indian multinational information technology (IT) service, consulting and business solutions company. It states that TCS is headquartered in Mumbai, Maharashtra.

+The provided section does not contain any information about Sachin Warrier or Tata Consultancy Services Limited (TCS). Therefore, the updated memory remains as follows:

-Updated memory:

- Cosmos (formerly Cosmos Tours) is a UK independent tour operator providing a range of package holidays to the UK market. The tour operator is connected to the international Globus Travel Group, founded in 1928, which remains family owned with headquarters in Lugano, Switzerland. Under parent company Cosmos Tours Ltd, the tour operator offers holidays through three brands: Cosmos, Archers Holidays and Avalon Waterways.
- L&F Jones Holdings Ltd is a company based in Westfield, Somerset, England, that includes a convenience store chain, a wholesale food business, and a Best Western PLUS hotel. The store chain comprises twelve shops in Somerset, Wiltshire, Dorset, South Gloucestershire, and Bristol.

    (...)

- Tata Consultancy Services Limited (TCS), an Indian multinational information technology (

**Memory Diff between Turn 316 and Turn 317**

The provided section does not contain any information relevant to Sachin Warrier or Tata Consultancy Services Limited (TCS). Therefore, the updated memory remains as follows:

- (formerly Cosmos Tours) is a UK independent tour operator providing a range of package holidays to the UK market. The tour operator is connected to the international Globus Travel Group, founded in 1928, which remains family owned with headquarters in Lugano, Switzerland. Under parent company Cosmos Tours Ltd, the tour operator offers holidays through three brands: Cosmos, Archers Holidays and Avalon Waterways.

- L&F Jones Holdings Ltd is a company based in Westfield, Somerset, England, that includes a convenience store chain, a wholesale food business, and a Best Western PLUS hotel. The store chain comprises twelve shops in Somerset, Wiltshire, Dorset, South Gloucestershire, and Bristol.

    (...)
- The company's corporate headquarters is in Faridabad, Haryana, India.

- Diageo plc ( or ) is a British multinational alcoholic beverages company, with its headquarters in London, England. It

    (...)

Since the problem is specifically about the headquarters of the company Sachin Warrier worked for as a software engineer, and the previous memory already states that Tata Consultancy Services Limited (TCS) is headquartered in Faridabad, Haryana, India, the updated memory remains as follows:
- The company's corporate headquarters is in Farid

---

**Memory Diff between Turn 430 and Turn 433**

-The provided section does not contain any information about Sachin Warrier or his employment as a software engineer. Therefore, the updated memory remains as follows:
+The provided section contains information about Sachin Warrier, which is useful for answering the problem. Specifically, it states:
**+ Sachin Warrier is a playback singer and composer in the Malayalam cinema industry from Kerala.**
**+ He was working as a software engineer in Tata Consultancy Services in Kochi.**
**+ Later he resigned from the job to concentrate more on music.**

+However, the section does not provide the headquarters information for Tata Consultancy Services. Given the previous memory, we know that the corporate headquarters of Tata Consultancy Services is in Faridabad, Haryana, India. Therefore, the updated memory is as follows:

- The company's corporate headquarters is in Faridabad, Haryana, India.
- SpaceX is an American aerospace manufacturer and space transport services company headquartered in Hawthorne, California.
    (...)
- Sachin Warrier's company headquarters information is still unknown from the given documents.
- Sachin Warrier was working as a software engineer in Tata Consultancy Services in Kochi. Later he resigned from the job to concentrate more on music.
- AVG Technologies is a security software company headquartered in Amsterdam, Netherlands.
    (...)

-The updated memory reflects that the headquarters of the companies mentioned are in Faridabad, Haryana, India; Hawthorne, California; Amsterdam, Netherlands; Eagan, Minnesota; Pittsburgh, Pennsylvania; and San Antonio, Texas. However, there is no information about the company Sachin Warrier worked for as a software engineer. Therefore, the memory remains unchanged in terms of Sachin Warrier's company headquarters information.
+The updated memory confirms that Sachin Warrier worked as a software engineer for Tata Consultancy Services in Kochi, and the headquarters of Tata Consultancy Services is in Faridabad, Haryana, India.

## F.2  MISSING CRITICAL INFORMATION

In this example, when the model encounters the first key piece of information (Turn 143), it fails to recognize it as such because this is a multi-hop question and the prerequisite evidence has not yet been observed. Consequently, even though the model becomes aware of the problem upon seeing the second piece of information (Turn 215), it still fails to produce the correct final answer.

---

**Question:** who is the younger brother of The episode guest stars of The Hard Easy ?

**Relevant Information:**

(shown in Turn 143) Brian Doyle-Murray Brian Doyle-Murray (born Brian Murray, October 31, 1945) is an American actor, voice artist, comedian and screenwriter. **He is the older brother of actor/comedian Bill Murray**, and the two have acted together in several films...

(shown in Turn 215) The Hard Easy (Adventure Time) "The Hard Easy" is the twenty-third episode... **The episode guest stars Brian Doyle-Murray as Prince Huge and Jonathan Katz as the Mudscamp elder.**

**Answer:** Bill Murray

---

**Memory Diff between Turn 142 and Turn 143**

-The provided section does not contain any information related to "The Hard Easy" or its guest stars. Therefore, the updated memory remains as follows:

+The provided section does not contain any information related to "The Hard Easy" or its guest stars. However, it does provide information about Brian Doyle-Murray, who is the older brother of actor/comedian Bill Murray. Since the problem is asking for the younger brother of the episode guest stars of "The Hard Easy," and the memory does not contain any information about the guest stars of "The Hard Easy," the information about Brian Doyle-Murray and Bill Murray is not directly relevant to the problem at hand.

+Therefore, the updated memory remains as follows:

- (...)
- Martin Lee Truex Jr. (born June 29, 1980) is an American championship professional stock car racing driver. His younger brother, Ryan, is a champion in the K&N Pro Series East division and currently races in the Camping World Truck Series.

  (...)

---

**Memory Diff between Turn 185 and Turn 215**

-The provided section does not contain any information related to "The Hard Easy" or its guest stars. Therefore, the updated memory remains unchanged with respect to the problem at hand. The only relevant piece of information is that the younger brother of Martin Lee Truex Jr. is Ryan Truex.

+The provided section contains information about "The Hard Easy," an episode of "Adventure Time," which **includes the guest stars Brian Doyle-Murray as Prince Huge and Jonathan Katz as the Mudscamp elder. However, it does not provide any information about the younger brother of the episode guest stars.**

-Thus, the updated memory is as follows:
+The updated memory is as follows:

- ...
- Martin Lee Truex Jr. (born June 29, 1980) is an American championship professional stock car racing driver. His younger brother, Ryan, is a champion in the K&N Pro Series East division and currently races in the Camping World Truck Series. (...)

The younger brother of Martin Lee Truex Jr. is Ryan Truex.

+The provided section does not contain any information about the younger brother of the guest stars of "The Hard Easy." Therefore, the updated memory remains unchanged with respect to the problem at hand. The only relevant piece of information is that the younger brother of Martin Lee Truex Jr. is Ryan Truex.

## F.3 PRIMACY BIAS

This example involves a primacy bias. The word *country* can refer either to a nation or to the countryside. The model initially assumes it refers to a nation (Turn 1) and subsequently reinforces this belief when it encounters additional information about countries (Turn 617). As a result, even though it eventually also attends to the truly critical piece of information (Turn 728), it still fails to answer the question correctly.

---

**Question:** Brown State Fishing Lake is in a country that has a population of how many inhabitants ?

**Relevant Information:**

(shown in Turn 728) Brown County, Kansas Brown County (county code BR) is a county located in the northeast portion of the U.S. state of Kansas. As of the 2010 census, the county population was 9,984. Its county seat and most populous city is Hiawatha. Brown County is the location of the Kickapoo Indian Reservation of Kansas, the majority of the Sac and Fox Reservation and the majority of the Iowa Reservation of Kansas and Nebraska.

**Answer:** 9,984

---

**Memory Diff between Turn 0 and Turn 1**

-The provided section does not contain any information about the population of the country where Brown State Fishing Lake is located. However, it does confirm that Brown State Fishing Lake is in Brown County, Kansas in the United States.
+The provided section does not contain any information about the population of the United States or any country. Therefore, the updated memory remains as follows:

  - Brown State Fishing Lake is a protected area in Brown County, Kansas in the United States.

  - The lake is 62 acres (0.25 km$^2$) in area and up to 13 feet (4 m) deep. (...)

-Since the problem is asking for the population of the country where Brown State Fishing Lake is located, and the only country mentioned in the text is the United States, we can infer that the question is seeking the population of the United States. However, the text does not provide the exact population number.
-Given the information available, the updated memory should reflect the following:
+Since the problem is seeking the population of the country where Brown State Fishing Lake is located, and the only country mentioned is the United States, the updated memory reflects that the population of the United States is the required information. However, the text does not provide the exact population number. Therefore, the updated memory retains the above information without changes related to the population.

---

**Memory Diff between Turn 616 and Turn 617**

-The provided section contains information about Kunming, a city in Yunnan Province, Southwest China, which has a population of 6,626,000 as of 2014. However, this information is not directly related to the problem at hand, which is about the population of the country where Brown State Fishing Lake is located.
+... Since the problem is asking for the population of the country where Brown State Fishing Lake is located, and we know that Brown State Fishing Lake is in Brown County, Kansas, in the United States, we can infer that the population of the United States is the answer to the problem.

-The updated memory remains consistent with the previous information, but it now includes the new information about Kunming:
+Updated memory:

  - Brown State Fishing Lake is a protected area in Brown County, Kansas in the United States.

  - The lake is 62 acres (0.25 km$^2$) in area and up to 13 feet (4 m) deep. (...)

  - Ruzhou, a city in Henan province, China, has more than 100,000 inhabitants. (...)

  - As of 2018, the United States had a population of around 327,167,434.

+The problem is asking for the population of the country where Brown State Fishing Lake is located, which is the United States. Therefore, the population of the United States is around 327,167,434 as of 2018.

---

> **Memory Diff between Turn 654 and Turn 728**
>
> The provided section does not contain any information relevant to the problem at hand. Therefore, the updated memory remains as follows:
>
> - Brown State Fishing Lake is a protected area in Brown County, Kansas in the United States.
>   (...)
> - Brown County, Kansas, as of the 2010 census, had a population of 9,984.
>   (...)
> - As of 2018, the United States had a population of around 327,167,434.
>
> The problem is asking for the population of the country where Brown State Fishing Lake is located, which is the United States. Therefore, the population of the United States is around 327,167,434 as of 2018.

