# OpenReview forum: "MemAgent: Reshaping Long-Context LLM with Multi-Conv RL-based Memory Agent"
_ICLR.cc/2026/Conference — ICLR 2026 Oral_

### Official Review · Reviewer_JGDV · 2025-10-30

**Soundness:** 3
**Presentation:** 4
**Contribution:** 4
**Rating:** 8
**Confidence:** 3

**Summary:**

The paper presents MEMAGENT, an agent that processes long documents chunk-by-chunk, using an RL-trained policy to update a fixed-size text buffer that serves as its memory. This sidesteps quadratic attention costs, achieving linear-time inference. The method demonstrates SOTA performance on several long-document QA benchmarks.

**Strengths:**

* This paper proposed a novel agent-based framework with clever memory design that gets excellent performance on long-document QA
* The approach reduce the complexity to linear-time inference complexity.

**Weaknesses:**

* The paper claims solving the problem of long-context LLM but mostly tested on synthetic or  semi-synthetic NIAH, RULER-HQA and extractive QA benchmarks. Form the memory architecture and prompt setup, this solution should well-suit the benchmarks given the major capability tested in these benchmarks are locating discrete facts. However, the same memory design may not work well with other tasks like summarization, etc.
* One major concerns regarding to the memory design is the error propagation. If there is a single step injecting error in the memory, it will propagate through the memory and finally affect the final results. More analysis on the failure patterns would make this draft more sound.
* The comparison is not sufficiently fair. The comparison mainly shows the performance diff between iterative-memory and single-pass attention. The experiment is hard to conclude that MEMAGENT is a "better" long-context model, but rather that an iterative approach is superior for these specific retrieval-heavy tasks. A more sound comparison would involve other iterative or memory-based techniques.

**Questions:**

* Can you provide a failure mode analysis, especially in the 3.5M token QA task. Does the distribution of errors related to the distance of required context?
* To better show the generalization of MemAgent, Could you provide results on a high-quality long-document summarization benchmark?
* The performance comparison is against an independent LLM out-of-box or fine-tuned. Could you discuss the performance comparison with other agent solution, too?

---

> ### Author Response · Authors · 2025-11-25
> **Rebuttal 1**
>
> Dear Reviewer JGDV,
>
> Thank you for recognizing the novelty and efficiency of our proposed method, as well as for your encouraging feedback. Below are our responses to the weaknesses and questions you raised:
>
> ## Q1 (and W2) Failure case study
>
> Thank you for pointing this out. During the rebuttal stage, we conduct a **comprehensive set of probing experiments** to examine MemAgent's behavior under different context distributions. Our findings show that, overall, **context distribution has only a mild impact on MemAgent's performance** (average variation under 3% across lengths). MemAgent remains robust and consistently strong even with the possibility of error propagation.
>
> In addition, **we provide several case studies from the 3.5M test set in Appendix F**, which you expressed interest in. We manually check the memory changing and highlight the core difference, and write a short summary for each sample. We hope these analyses offer further intuitive insight into how MemAgent behaves in challenging scenarios.
> - **Failure in Writing: Information Overwritten**: In this example, the model accumulates irrelevant memories in the early stage (Turn 60). When the crucial context appears, the model does capture the relevant information (Turn 289), but attempts to append it to the end (Turn 290), where it is truncated due to insufficient memory. Later, the model performs a summarization (Turn 317), which gives it more space to store the second key piece of context (Turn 433). However, since the information from the first context has already been lost, the model still fails to answer the question correctly.
> - **Failure in Reading: Missing Key Information**: In this example, when the model encounters the first key piece of information (Turn 143), it fails to recognize it as such because this is a multi-hop question and the prerequisite evidence has not yet been observed. Consequently, even though the model becomes aware of the problem upon seeing the second piece of information (Turn 215), it still fails to produce the correct final answer.
> - **Failure in Reasoning: Primacy Bias** The word *country* can refer either to a nation or to the countryside. The model initially assumes it refers to a nation (Turn 1) and subsequently reinforces this belief when it encounters additional information about countries (Turn 617). As a result, even though it eventually also attends to the truly critical piece of information (Turn 728), it still fails to answer the question correctly.
>
> > **Probing experiment results.** random indicates randomly shuffling all context items, consistent with the setup in the main experiment. The other rows show the performance difference relative to random. Ctx. Dist. denotes the context distribution, where the two numbers correspond to the relative positions of the two key-information groups within the entire context. 0% means the beginning and 100% means the end.
>
> | Model | Ctx. Dist.   | 7K      | 14K       | 28K       | 56K       | 112K      | 224K      | 448K       | AVG       |
> |--|---|---|--|--|--|-|--|--|--|
> | 14B   | random       | +80.47  | +82.03    | +82.03    | +83.59    | +81.25    | +77.34    | +79.69     | +75.78    |
> |     | 0% 20%   | +3.91   | -3.91     | +3.13     | +1.57     | 0.00      | +4.69     | +3.90      | +1.90     |
> |       | 0% 100%   | +3.12   | +0.78     | +3.13     | -3.12     | +1.56     | +7.82     | +6.25      | +2.79     |
> |    | 20% 80%   | +0.78   | -3.12     | +2.35     | +0.79     | -3.13     | +5.47     | -3.13      | 0.00      |
> |     | 40% 60%   | +1.56   | +2.35     | -2.34     | -3.12     | -1.56     | +3.13     | -0.78      | -0.11     |
> |    | 80% 100%  | -2.35   | 0.00      | +1.56     | +0.79     | +3.13     | 0.00      | +1.56      | +0.67     |
> | 7B    | random    | +81.25  | +81.25    | +82.03    | +80.47    | +79.69    | +75.78    | +76.56     | +79.58    |
> |       | 0% 20%    | -1.56   | -0.78     | +3.13     | +3.91     | +3.12     | +5.47     | +3.13      | +2.35     |
> |    | 0% 100%      | +0.78   | 0.00      | +2.35     | +1.56     | +3.90     | +4.69     | +2.35      | +2.23     |
> |    | 20% 80%      | -0.78   | -0.78     | +2.35     | 0.00      | 0.00      | 0.00      | +3.13      | +0.56     |
> |       | 40% 60%      | 0.00    | +1.56     | +3.13     | +0.78     | -3.91     | +3.13     | +5.47      | +1.45     |
> |       | 80% 100%     | +1.56   | 0.00      | +0.78     | -0.78     | +0.78     | +4.69     | 0.00       | +1.00     |

---

> ### Author Response · Authors · 2025-11-25
> **Rebuttal 2**
>
> ## Q2（and W1）Summary Tasks
>
> Thank you for asking this. We conduct experiments on two summarization tasks from LongBench—**GOVREPORT[1]** and **QMSUM[2]**. The results show that **MemAgent naturally generalizes to summarization tasks without any additional training**.
>
> > Comparison of RL-MemAgent and baseline models on summary benchmarks. All values represent recall rates (%). **Bold** = Highest value, *Italic* = Second highest value in each column.
>
> | Model                      | | |GOV REPORT | | | |QMSUM | |
> |----------------------------|-----------------|-----------------|-----------------|-----------------|-----------------|-----------------|-----------------|-----------------|
> |                            | ROUGE-1         | ROUGE-2         | ROUGE-L         | AVG             | ROUGE-1         | ROUGE-2         | ROUGE-L         | AVG             |
> | Qwen2.5-Instruct-32B       | 23.67           | 8.46            | 12.57           | 14.90           | 47.77           | 11.29           | 28.17           | 29.08           |
> | Qwen2.5-Instruct-14B       | 31.19           | 10.96           | 14.96           | 19.04           | 47.53           | 11.46           | 28.28           | 29.09           |
> | Qwen2.5-Instruct-7B        | 30.91           | 11.68           | 15.20           | 19.26           | 46.64           | 12.01           | 28.33           | 28.99           |
> | QwenLong-L1                | 27.60           | 8.20            | 13.07           | 16.29           | 39.44           | 8.24            | 23.56           | 23.74           |
> | Qwen2.5-Instruct-14B-1M    | 30.58           | 11.93           | 15.51           | *19.34*    | 47.31           | 13.13           | 29.07           | *29.84*    |
> | Qwen2.5-Instruct-7B-1M     | 31.02           | 11.47           | 15.30           | 19.26           | 46.72           | 12.33           | 28.66           | 29.24           |
> | DS-Distill-Qwen-32B        | 26.13           | 8.86            | 12.98           | 15.99           | 39.09           | 8.75            | 23.96           | 23.93           |
> | DS-Distill-Qwen-14B        | 28.24           | 9.72            | 13.78           | 17.25           | 41.25           | 8.95            | 25.00           | 25.07           |
> | DS-Distill-Qwen-7B         | 33.30           | 9.39            | 14.59           | 19.10           | 34.33           | 5.97            | 21.57           | 20.62           |
> | **RL-MemAgent-14B**       | **37.16**       | *12.03*    | **16.23**       | **21.80**       | **50.21**       | *12.70*    | **31.27**       | **31.39**       |
> | **RL-MemAgent-7B**        | 30.28           | **12.37**       | *15.37*    | *19.34*    | *48.49*    | **14.41**       | *30.91*    | *31.27*    |

---

> ### Author Response · Authors · 2025-11-25
> **Rebuttal 3**
>
> ## Q3 (and W3) Agent Baselines
>
> Thank you for pointing this out. During the rebuttal stage, we compare MemAgent against an advanced memory-agent method, **Mem0** [3]. The Mem0 paper also reports that RAG methods using only top-1 or top-2 retrieval form strong and stable baselines for memory-agent tasks. Therefore, we conduct extensive comparisons against RAG agents under multiple configurations.
>
> The results show that **MemAgent outperforms these methods, demonstrating that end-to-end RL–trained memory provides greater flexibility and coherence compared with retrieval-based strategies.**
>
> For Mem0, we use SOTA OpenAI models, GPT-5.1 and text-embedding-3-large as langugae model and embedding model respectively and we follow the official GitHub repository for memory updating and retrieval. Specifically, during memory creation, we split and processed the entire context in 5,000-token chunks; during retrieval, we selected the top 30 memories.
>
> Although we have tuned the PE, Mem0 still fails to achieve strong performance—likely due to a mismatch between its intended application scenarios and our task. This does not diminish Mem0's strong capabilities in chatbot settings, but highlights its limitations when applied to long-document understanding.
>
> | **Method**       | **2Wiki** | **HQA** | **MuSiQue** | **NQA** | **Qasper** | **RULER-HQA-7K** | **RULER-HQA-14K** | **RULER-HQA-28K** |
> |------------------|-----------|---------|-------------|---------|-----------|------------------|-------------------|-------------------|
> | RL-MemAgent-7B   | 74.00      | 69.50    | 47.00        | 21.50    | 29.00      | 81.25             | 81.25             | 82.03             |
> | Mem0             | 65.50     | 61.50   | 40.00       | 24.50   | 15.00     | 49.22            | 43.75             | 37.50             |
>
> For RAG Agent, we also use text-embedding-3-large as embedding model.
>
> ## Reference
> > [1] Huang L, Cao S, Parulian N, Ji H, Wang L. Efficient attentions for long document summarization. arXiv preprint arXiv:2104.02112, 2021.
>
> > [2] Zhong M, Yin D, Yu T, Zaidi A, Mutuma M, Jha R, Hassan A, Celikyilmaz A, Liu Y, Qiu X, et al. QMSum: A new benchmark for query-based multi-domain meeting summarization. In: Proceedings of the 2021 Conference of the North American Chapter of the Association for Computational Linguistics: Human Language Technologies, 2021: 5905-5921.
>
> > [3] Chhikara P, Khant D, Aryan S, Singh T, Yadav D. Mem0: Building production-ready ai agents with scalable long-term memory. arXiv preprint arXiv:2504.19413, 2025.

---

> ### Author Response · Authors · 2025-11-25
> **Rebuttal 4**
>
> ## Q3 (and W3): Experiments Results
> > Result versus RAG Agent in RULER-HQA with different top-K settings. We segment the context based on natural semantic units, i.e., each wiki item was treated as a chunk.
>
> | Model               | 7K     | 14K    | 28K    | 56K    | 112K   | 224K   | 448K   | 896K   |
> |---------------------|--------|--------|--------|--------|--------|--------|--------|--------|
> | RAG + Qwen2.5-14B   |        |        |        |        |        |        |        |        |
> | *K=2*               | 57.03  | 54.69  | 51.56  | 54.69  | 53.12  | 50.00  | 52.34  | 49.22  |
> | *K=4*               | 66.41  | 67.19  | 68.75  | 67.19  | 66.41  | 64.06  | 66.41  | 64.84  |
> | *K=6*               | 72.66  | 75.78  | 75.78  | 74.22  | 69.53  | 71.88  | 73.44  | 67.19  |
> | *K=8*               | 78.12 | 78.91 | 77.34 | 81.25 | 76.56 | 78.12 | 77.34 | 74.22 |
> | **RL-MemAgent-14B** | **80.47** | **82.03** | **82.03** | **83.59** | **81.25** | **77.34** | **79.69** | **75.78** |
> | RAG + Qwen2.5-7B    |        |        |        |        |        |        |        |        |
> | *K=2*               | 53.91  | 54.69  | 53.12  | 51.56  | 54.69  | 51.56  | 52.34  | 49.22  |
> | *K=4*               | 67.19  | 66.41  | 66.41  | 67.19  | 64.84  | 64.06  | 62.50  | 61.72  |
> | *K=6*               | 74.22  | 73.44  | 72.66  | 73.44  | 470.31 | 73.44  | 70.31  | 67.19  |
> | *K=8*               | 75.00 | 75.00 | 75.78 | 74.22 | 74.22 | 77.34 | 72.66 | 68.75 |
> | **RL-MemAgent-7B**  | **81.25** | **81.25** | **82.03** | **80.47** | **79.69** | **75.78** | **76.56** | **74.22** |
>
> > Result versus RAG Agent in Longbench-QA with different top-K and Context size settings. We segment the context using fixed-length chunks. For retrieval, we performed top-k matching using cosine similarity scores.
>
> | **Method**       | **2Wiki** | **HQA** | **MuSiQue** | **NQA** | **Qasper** | **AVG** |
> |----------------|--------|--------|--------|--------|--------|--------|
> |Qwen2.5-7B + RAG |   |   |   |   |   |   |
> |*C=1024 K=2* | 41.00 | 48.50 | 22.00 | 14.50 | 25.50 | 25.25 |
> |*C=1024 K=4* | 49.00 | 56.50 | 28.00 | 17.00 | 28.50 | 29.83 |
> |*C=1024 K=6* | 54.50 | 57.50 | 29.50 | 17.00 | 25.00 | 30.58 |
> |*C=1024 K=8* | 50.50 | 59.00 | 29.50 | 18.00 | 25.00 | 30.33 |
> |*C=2048 K=2* | 49.50 | 51.50 | 19.50 | 12.50 | 27.00 | 26.67 |
> |*C=2048 K=4* | 50.50 | 53.00 | 26.00 | 17.00 | 25.50 | 28.67 |
> |*C=2048 K=6* | 50.50 | 56.50 | 27.50 | **22.00** | 25.50 | 30.33 |
> |*C=2048 K=8* | 50.50 | 58.00 | 25.50 | 21.00 | 27.00 | 30.33 |
> |**RL-MemAgent-7B** | **74.00** | **69.50** | **47.00** | 21.50 | **29.00** | **48.20** |
> |Qwen2.5-14B + RAG |   |   |   |   |   |   |
> |*C=1024 K=2* | 51.50 | 56.50 | 26.50 | 15.00 | 23.50 | 28.83 |
> |*C=1024 K=4* | 70.00 | 64.50 | 34.50 | 17.50 | 27.00 | 35.58 |
> |*C=1024 K=6* | 71.50 | 64.00 | 41.00 | 19.00 | **27.50** | 37.17 |
> |*C=1024 K=8* | 72.50 | 64.50 | 39.00 | 17.50 | 26.00 | 36.58 |
> |*C=2048 K=2* | 58.50 | 61.50 | 33.50 | 13.50 | 25.50 | 32.08 |
> |*C=2048 K=4* | 76.00 | 64.00 | 36.00 | 18.50 | 25.00 | 36.58 |
> |*C=2048 K=6* | 73.00 | 67.50 | 41.50 | 21.00 | 26.50 | 38.25 |
> |*C=2048 K=8* | 77.50 | 68.50 | 42.00 | 21.00 | **27.50** | 39.42 |
> |**RL-MemAgent-14B** | **79.0** | **73.0** | **52.00** | **25.00** | 26.00 | **51.00** |

---

### Official Review · Reviewer_BZGs · 2025-10-31

**Soundness:** 4
**Presentation:** 4
**Contribution:** 4
**Rating:** 8
**Confidence:** 4

**Summary:**

The paper introduces MemAgent, a novel agent-based workflow designed to allow LLMs to process arbitrarily long documents with linear time-complexity and minimal performance loss. Instead of processing an entire document at once, MemAgent divides the text into smaller chunks. It then iteratively reads each chunk alongside a fixed-length "memory" buffer. After each chunk, the model uses a simple "overwrite" strategy to update this memory with what it deems important.The core innovation is in its training.

The model is trained using a new Reinforcement Learning (RL) algorithm called Multi-conv DAPO. This method treats the entire sequence of memory updates for a single document as one long trajectory. A reward is given only for the final answer's correctness, and this reward is used to optimize all preceding memory-writing decisions. Experiments show that this approach is highly effective. A model trained with an 8K context window (using a 5K chunk and 1K memory) can extrapolate to answer questions on documents as long as 3.5 million tokens with less than 10% performance degradation—a task where standard long-context models fail completely.

**Strengths:**

1. **Strong Performance:** Extrapolating from an 8K training context to 3.5M tokens shows only minimal performance loss,  effectively solves the crucial problem in long-context modeling. The strong results on RULER-HQA, LongBench-QA, and NIAH demonstrate state-of-the-art performance and generalization.
2. **Novel and Effective RL Framework:** The Multi-conv DAPO algorithm is a clever solution to a difficult credit-assignment problem. By propagating the final answer's reward back to all intermediate memory-update steps, the model is directly optimized to learn what to remember to succeed at the final task. This shows an easy way to improve current model's ability.
3. **Well written paper**: The paper is well written and easy to follow.

**Weaknesses:**

1. Memory management seems to be a new lost-in-the-middle: In the ablation shown in Figure 9, increasing memory size does not seem to improve performance but rather decrease, and so this still demonstrate the model inherent problem with handling long-context (in this case the context window of the memory).
2. The memory length is different for different tasks, and this may seem an additional hyper-parameter which may be costly to tune.

**Questions:**

1. Do the authors have any intuition how this may be adapted to other tasks, like long-form summarization, where there is no verifiable outcome reward?
2. I am wondering if the authors have explored the lost-in-the-middle for MeMAgent, i.e. do the models still perform worse when the crucial information is in the middle or can this strategy effectively fix this problem?

---

> ### Author Response · Authors · 2025-11-25
> **Rebuttal 1**
>
> Dear Reviewer BZGs,
>
> Thank you for recognizing the performance of our MemAgent, the novelty of our RL training framework, and for your encouraging feedback. Below are our responses to the weaknesses and questions you raised:
>
> ## W1 & W2: Memory Size
>
> Thank you for pointing this out. We conduct a new set of ablation studies, which shows that, **although changes in memory size may slightly affect performance across different tasks, the overall fluctuation is minor.**
>
> This indicates that expensive hyperparameter tuning is not necessary and demonstrates the robustness of MemAgent. In practice, it is sufficient to maintain the same configuration used during training. While memory scaling may not behave as smoothly as reasoning scaling, this is likely due to the nature of long-context tasks.
>
> In the new experiments, we keep the total length of *memory size + context chunk size* constant, and linearly adjusted the memory size. The goal is to control the total context length per conversation turn.
>
> - In the original ablation study, increasing the memory size results in differences of less than 1% on LongBench average scores and less than 5% on NIAH.
> - In the new supplementary ablation, differences remains below 3% on LongBench and below 2% on NIAH.
> - In the original ablation, performance fluctuations appear only when the memory size was heavily compressed (down to 1/4 of the original, i.e., 256 tokens, where the model can store only a few sentences). Even in this extreme case, the performance drop remains below 3% on LongBench and below 8% on NIAH. Given how unrealistically small the memory is in this setting, we believe these results further demonstrate MemAgent's robustness with respect to the memory-size parameter.
> - Combining both sets of experiments, we observe that none of the hyperparameter combinations strongly affect the performance of the memory agent, and the best configuration is simply the training configuration. This effectively resolves the issue of tuning memory-related hyperparameters.
>
>
> > Ablation result of memory-size and context-size on NIAH
>
>  | Method     | **8K**    | **16K**   | **32K**   | **64K**   | **128K**  | **256K**  | **512K**  |
> |------------|-------|-------|-------|-------|-------|-------|-------|
> | 14B-m4096-c1928  | 100.00 | 99.22 | 98.70 | 97.66 | 98.96 | 99.22 | 97.92 |
> | 14B-m3072-c2952  | 99.74 | 100.00 | 100.00 | 100.00 | 98.70 | 99.48 | 99.48 |
> | 14B-m2048-c3976  | 99.74 | 99.48 | 99.48 | 100.00 | 100.00 | 97.13 | 97.66 |
> | 14B-m1024-c5000  | 99.74 | 98.70 | 100.00 | 100.00 | 98.96 | 97.40 | 98.18 |
> | 7B-m4096-c1928  | 100.00 | 99.22 | 98.44 | 96.35 | 98.96 | 98.18 | 97.14 |
> | 7B-m3072-c2952  | 99.48 | 100.00 | 99.74 | 99.48 | 96.88 | 98.18 | 96.09 |
> | 7B-m2048-c3976  | 99.74 | 99.48 | 99.48 | 99.48 | 98.96 | 94.53 | 94.53 |
> | 7B-m1024-c5000  | 100.00 | 98.70 | 99.74 | 98.96 | 97.40 | 97.66 | 96.62 |
>
> > Ablation result of memory-size and context-size on Longbench-QA
>
> | Dataset    | **2Wiki** | **HQA** | **MuSiQue** | **NQA** | **Qasper** | **AVG** |
> |------------|-----------|---------|-------------|---------|------------|---------|
> | 14B-m4096-c1928  | 74.5      | 72.5    | 48.5        | 21.5    | 25.5       | 48.5    |
> | 14B-m3072-c2952  | 76.5      | 70.5    | 52.5        | 24.5    | 26.5       | 50.1    |
> | 14B-m2048-c3976  | 74.5      | 71.5    | 49.5        | 23.0    | 27.0       | 49.1    |
> | 14B-m1024-c5000  | 79.0      | 73.0    | 52.0        | 25.0    | 26.0       | 51.0    |
> | 7B-m4096-c1928  | 70.0      | 66.0    | 45.5        | 19.0    | 26.0       | 45.3    |
> | 7B-m3072-c2952  | 72.0      | 64.0    | 44.0        | 20.0    | 25.5       | 45.1    |
> | 7B-m2048-c3976  | 75.0      | 69.0    | 43.5        | 23.0    | 26.0       | 47.3    |
> | 7B-m1024-c5000  | 74.0      | 69.5    | 47.0        | 21.5    | 29.0       | 48.2    |

---

> ### Author Response · Authors · 2025-11-25
> **Rebuttal 2**
>
> ## Q1: Summary Tasks
> Thank you for asking this. Our goal for MemAgent's training is to develop a **general memory capability**: extracting key information, retaining it, organizing it effectively, and then leveraging it appropriately during question answering. In fact, we find that **MemAgent can generalize to long-document summarization tasks without any additional training**. Automatic metrics such as ROUGE could also serve as outcome rewards in RLVR if needed.
>
>
> > Comparison of RL-MemAgent and baseline models on summary benchmarks. All values represent recall rates (%). **Bold** = Highest value, *Italic* = Second highest value in each column.
>
> | Model                      | | |**GOV REPORT**[1] | | | |**QMSUM**[2] | |
> |----------------------------|-----------------|-----------------|-----------------|-----------------|-----------------|-----------------|-----------------|-----------------|
> |                            | ROUGE-1         | ROUGE-2         | ROUGE-L         | AVG             | ROUGE-1         | ROUGE-2         | ROUGE-L         | AVG             |
> | Qwen2.5-Instruct-32B       | 23.67           | 8.46            | 12.57           | 14.90           | 47.77           | 11.29           | 28.17           | 29.08           |
> | Qwen2.5-Instruct-14B       | 31.19           | 10.96           | 14.96           | 19.04           | 47.53           | 11.46           | 28.28           | 29.09           |
> | Qwen2.5-Instruct-7B        | 30.91           | 11.68           | 15.20           | 19.26           | 46.64           | 12.01           | 28.33           | 28.99           |
> | QwenLong-L1                | 27.60           | 8.20            | 13.07           | 16.29           | 39.44           | 8.24            | 23.56           | 23.74           |
> | Qwen2.5-Instruct-14B-1M    | 30.58           | 11.93           | 15.51           | *19.34*    | 47.31           | 13.13           | 29.07           | *29.84*    |
> | Qwen2.5-Instruct-7B-1M     | 31.02           | 11.47           | 15.30           | 19.26           | 46.72           | 12.33           | 28.66           | 29.24           |
> | DS-Distill-Qwen-32B        | 26.13           | 8.86            | 12.98           | 15.99           | 39.09           | 8.75            | 23.96           | 23.93           |
> | DS-Distill-Qwen-14B        | 28.24           | 9.72            | 13.78           | 17.25           | 41.25           | 8.95            | 25.00           | 25.07           |
> | DS-Distill-Qwen-7B         | 33.30           | 9.39            | 14.59           | 19.10           | 34.33           | 5.97            | 21.57           | 20.62           |
> | **RL-MemAgent-14B**       | **37.16**       | *12.03*    | **16.23**       | **21.80**       | **50.21**       | *12.70*    | **31.27**       | **31.39**       |
> | **RL-MemAgent-7B**        | 30.28           | **12.37**       | *15.37*    | *19.34*    | *48.49*    | **14.41**       | *30.91*    | *31.27*    |

---

> ### Author Response · Authors · 2025-11-25
> **Rebuttal 3**
>
> ## Q2: Lost-in-the-middle
>
> Thank you for pointing this out. During the rebuttal stage, we conduct a comprehensive set of probing experiments to examine MemAgent's behavior under different context distributions. The conclusion is that **MemAgent exhibits almost no lost-in-the-middle phenomenon**. Even when the key information is placed at the 40% and 60% positions, the performance drop relative to the random distribution baseline is only 0.11% on average. **It is true, however, that the model is more sensitive to key information appearing at the very beginning or very end of the context.**
>
> Overall, MemAgent remains stable across a wide range of context distributions. This robustness arises because, during training, we randomized the positions of the key information within the full context, enabling the model to learn a strategy that is near-optimal across all patterns.
>
> > **Probing experiment results.** "random" indicates randomly shuffling all context items, consistent with the setup in the main experiment. The other rows show the performance difference relative to "random." "Ctx. Dist." denotes the context distribution, where the two numbers correspond to the relative positions of the two key-information groups within the entire context. 0% means the beginning and 100% means the end.
>
> | Model | Ctx. Dist.   | 7K      | 14K       | 28K       | 56K       | 112K      | 224K      | 448K       | AVG       |
> |-------|--------------|---------|-----------|-----------|-----------|-----------|-----------|------------|-----------|
> | 14B   | random       | +80.47  | +82.03    | +82.03    | +83.59    | +81.25    | +77.34    | +79.69     | +75.78    |
> |       | 0% 20%       | +3.91   | -3.91     | +3.13     | +1.57     | 0.00      | +4.69     | +3.90      | +1.90     |
> |       | 0% 100%      | +3.12   | +0.78     | +3.13     | -3.12     | +1.56     | +7.82     | +6.25      | +2.79     |
> |       | 20% 80%      | +0.78   | -3.12     | +2.35     | +0.79     | -3.13     | +5.47     | -3.13      | 0.00      |
> |       | 40% 60%      | +1.56   | +2.35     | -2.34     | -3.12     | -1.56     | +3.13     | -0.78      | -0.11     |
> |       | 80% 100%     | -2.35   | 0.00      | +1.56     | +0.79     | +3.13     | 0.00      | +1.56      | +0.67     |
> | 7B    | random       | +81.25  | +81.25    | +82.03    | +80.47    | +79.69    | +75.78    | +76.56     | +79.58    |
> |       | 0% 20%       | -1.56   | -0.78     | +3.13     | +3.91     | +3.12     | +5.47     | +3.13      | +2.35     |
> |       | 0% 100%      | +0.78   | 0.00      | +2.35     | +1.56     | +3.90     | +4.69     | +2.35      | +2.23     |
> |       | 20% 80%      | -0.78   | -0.78     | +2.35     | 0.00      | 0.00      | 0.00      | +3.13      | +0.56     |
> |       | 40% 60%      | 0.00    | +1.56     | +3.13     | +0.78     | -3.91     | +3.13     | +5.47      | +1.45     |
> |       | 80% 100%     | +1.56   | 0.00      | +0.78     | -0.78     | +0.78     | +4.69     | 0.00       | +1.00     |
>
> ## Reference
> > [1] Huang L, Cao S, Parulian N, Ji H, Wang L. Efficient attentions for long document summarization. arXiv preprint arXiv:2104.02112, 2021.
>
> > [2] Zhong M, Yin D, Yu T, Zaidi A, Mutuma M, Jha R, Hassan A, Celikyilmaz A, Liu Y, Qiu X, et al. QMSum: A new benchmark for query-based multi-domain meeting summarization. In: Proceedings of the 2021 Conference of the North American Chapter of the Association for Computational Linguistics: Human Language Technologies, 2021: 5905-5921.

---

> > ### Comment · Reviewer_BZGs · 2025-11-26
> >
> > Thank you for the detailed reply and extensive additional results! I am curious about two points:
> > 1. For summarization setting, since there is no real verifiable rewards as rouge score is not very a perfect and reliable to represent "true" summarization results, have you noticed any reward hacking occuring? For example, what would happen if the final evaluation of the metrics are performed via LLM-as-a-judge. Does a new automatic metrics still show the same trend that RL-MemAgent performs the best? I ask this primarily that in my experiments long time ago models learned to just copy from the document in order to achieve high Rouge scores rather than truely generating good summaries.
> > 2. The list-in-the-middle experiments are very good! Do you think the reason RL-MemAgent is robust is more because the training contains mixed positions so that the model learns not to be biased or more in the architecture of forcing it to always keep good information in the memory?

---

> ### Author Response · Authors · 2025-11-28
>
> We appreciate your timely response and recognition of our additional experiments! Below are our answers to the issues you raised.
>
> ## Q1: Reward Hacking
>
> We manually examined some cases and found that **reward hacking is rare across various models, especially when they are not post-trained on it**. A simple approach to quantify this behavior is to calculate **the length of the longest (continuous) common substring** between the original content and the summary generated by the LLM, and compare it with the reference summary. Our results (measured in characters) show that **the model produces summaries with minimal repetition of the original content.**
>
> > Length of the longest common substring (in characters) between the document and reference/model's summary in different datasets.
> |**Summary** | **GovReport** | **QMSum**|
>  | --|--|--|
> |Reference |84|7.19|
> |MemAgent-7B |131|10.69|
>  |MemAgent-14B |117|6.155|
>
> LLM-as-a-judge exhibits some peculiar preferences, such as generally favoring responses generated by its own model family and tending to assign higher scores to structured text rather than natural paragraphs. **Training a model to effectively critique and evaluate open-ended responses based on certain principles (referred to as rubric RL) remains a challenge.** This can be an important future enhancemen for the existed RLVR paradigm.
>
> ## Q2: Robustness over Various Context Distribution
>
> Thank you for your question! We believe that **both factors play a role in this context.** First, the MemAgent architecture treats memory processing as a series of **Markov processes: past mistakes cannot be undone, but as long as caution is exercised, no further historical interference occurs.** Secondly, when the model is trained on a uniformly distributed dataset, it learns **the art of balancing**: recording relevant information early is beneficial because the context for multiple inferences may reappear later; appropriately forgetting irrelevant content helps conserve memory space; and most importantly, once key information is identified, it must be retained throughout. **These two factors together contribute to MemAgent's robustness.**

---

### Official Review · Reviewer_BjEW · 2025-11-01

**Soundness:** 2
**Presentation:** 3
**Contribution:** 3
**Rating:** 6
**Confidence:** 4

**Summary:**

This paper introduces MEMAGENT, a novel agent-based framework designed to enable LLMs to process arbitrarily long contexts with linear time complexity. The core idea is to process a long document in discrete segments. The model maintains a fixed-size "memory" buffer within its context window, which it iteratively updates after processing each segment. This overwrite strategy avoids the quadratic complexity of standard attention mechanisms.

The key contribution is the training methodology. The authors formulate the memory update mechanism as a policy to be learned through RL. They propose Multi-Conv DAPO, an extension of DAPO, to optimize the entire multi-step agent workflow based on a final outcome reward. In experiments, a model with an 8K context window trained using the MEMAGENT workflow demonstrates remarkable extrapolation capabilities, successfully handling QA tasks on documents up to 3.5 million tokens with minimal performance degradation, significantly outperforming existing long-context models.

**Strengths:**

1. The agent-based memory workflow is an elegant and practical solution to the long-context problem, sidestepping the quadratic complexity of attention.
2. The experimental results are outstanding. The ability to extrapolate from an 8K training context to a 3.5M token QA task with less than 10% performance drop is great.
3. By design, the method scales linearly with the length of the input document in terms of both time and memory, making it highly efficient for real-world deployment on extremely long texts.
4. The paper successfully demonstrates that RL can be used to train an LLM to perform a complex, multi-step reasoning task like dynamic memory management, with the proposed Multi-Conv DAPO algorithm being a key enabler.

**Weaknesses:**

1. The fixed-size memory is the source of the method's efficiency, but it's also a potential bottleneck. For tasks that require synthesizing many disparate pieces of information from across a long document, the model might discard critical information prematurely. The paper could discuss this trade-off more explicitly and analyze failure cases where this occurs. Moreover, I think a better choice is to use variable-sized memory based on the amount of information contained in the context. How to implement this in the current framework should be explained and experimented with, if possible.
2. The evaluation is heavily focused on extractive QA. It is unclear how well this memory management strategy would generalize to other long-context tasks like summarization, multi-document comparison, or creative writing, which may require different memory retention patterns.
3. The RL-based training setup is sophisticated, involving a custom algorithm (Multi-Conv DAPO) and multiple stages. This poses a high barrier to reproduction for researchers without access to similar frameworks and expertise. More detailed pseudocode or implementation guidance would be beneficial.
4. The choice to apply the final advantage value uniformly to all preceding memory-update steps is a simple credit assignment strategy. This could be suboptimal, as some memory updates are likely more critical than others. This represents a potential area for future improvement.

**Questions:**

Section 2.3 presents a theoretical latent-variable view of the process. How does this formal interpretation connect to the practical implementation of the RL agent? Is the agent learning to approximate a posterior distribution over memory states, or is it learning a more deterministic policy to generate a single memory trajectory?

---

> ### Author Response · Authors · 2025-11-25
> **Rebuttal 1**
>
> Dear Reviewer BjEW,
>
> Thank you for recognizing the effectiveness of our MemAgent and the Multi-Conv DAPO algorithm and your supportive feedback. Below are our responses to the weaknesses and questions you raised:
>
> ## W1.1 Failure cases of discarding critical information
>
> Thank you for pointing this out. We conduct detailed **probing experiments** to analyze how the model behaves under different context distributions, and we also add **case studies in Appendix F** to illustrate representative failure cases.
> What we aim to demonstrate is that **the model does not suffer catastrophic degradation even under various extreme patterns.**
>
> > **Probing experiment results.** "random'' indicates randomly shuffling all context items, consistent with the setup in the main experiment. The other rows show the performance difference relative to "random.'' "Ctx. Dist.'' denotes the context distribution, where the two numbers correspond to the relative positions of the two key-information groups within the entire context. 0% means the beginning and 100% means the end.
>
> | Model | Ctx. Dist.   | 7K      | 14K       | 28K       | 56K       | 112K      | 224K      | 448K       | AVG       |
> |-------|--------------|---------|-----------|-----------|-----------|-----------|-----------|------------|-----------|
> | 14B   | random       | +80.47  | +82.03    | +82.03    | +83.59    | +81.25    | +77.34    | +79.69     | +75.78    |
> |       | 0% 20%       | +3.91   | -3.91     | +3.13     | +1.57     | 0.00      | +4.69     | +3.90      | +1.90     |
> |       | 0% 100%      | +3.12   | +0.78     | +3.13     | -3.12     | +1.56     | +7.82     | +6.25      | +2.79     |
> |       | 20% 80%      | +0.78   | -3.12     | +2.35     | +0.79     | -3.13     | +5.47     | -3.13      | 0.00      |
> |       | 40% 60%      | +1.56   | +2.35     | -2.34     | -3.12     | -1.56     | +3.13     | -0.78      | -0.11     |
> |       | 80% 100%     | -2.35   | 0.00      | +1.56     | +0.79     | +3.13     | 0.00      | +1.56      | +0.67     |
> | 7B    | random       | +81.25  | +81.25    | +82.03    | +80.47    | +79.69    | +75.78    | +76.56     | +79.58    |
> |       | 0% 20%       | -1.56   | -0.78     | +3.13     | +3.91     | +3.12     | +5.47     | +3.13      | +2.35     |
> |       | 0% 100%      | +0.78   | 0.00      | +2.35     | +1.56     | +3.90     | +4.69     | +2.35      | +2.23     |
> |       | 20% 80%      | -0.78   | -0.78     | +2.35     | 0.00      | 0.00      | 0.00      | +3.13      | +0.56     |
> |       | 40% 60%      | 0.00    | +1.56     | +3.13     | +0.78     | -3.91     | +3.13     | +5.47      | +1.45     |
> |       | 80% 100%     | +1.56   | 0.00      | +0.78     | -0.78     | +0.78     | +4.69     | 0.00       | +1.00     |
>
> For case study, we manually check the memory changing and highlight the core difference, and write a short summary for each sample. We hope these analyses offer further intuitive insight into how MemAgent behaves in challenging scenarios.
> - **Failure in Writing: Information Overwritten**: In this example, the model accumulates a large amount of irrelevant memories in the early stage (Turn 60). When the crucial context appears, the model does capture the relevant information (Turn 289), but attempts to append it to the end (Turn 290), where it is truncated due to insufficient memory. Later, the model proactively performs a summarization (Turn 317), which gives it more space to store the second key piece of context (Turn 433). However, since the information from the first context has already been lost, the model incorrectly relies on irrelevant information for reasoning and fails to answer the question correctly.
> - **Failure in Reading: Missing Key Information**: In this example, when the model encounters the first key piece of information (Turn 143), it fails to recognize it as such because this is a multi-hop question and the prerequisite evidence has not yet been observed. Consequently, even though the model becomes aware of the problem upon seeing the second piece of information (Turn 215), it still fails to produce the correct final answer.
> - **Failure in Reasoning: Primacy Bias** The word *country* can refer either to a nation or to the countryside. The model initially assumes it refers to a nation (Turn 1) and subsequently reinforces this belief when it encounters additional information about countries (Turn 617). As a result, even though it eventually also attends to the truly critical piece of information (Turn 728), it still fails to answer the question correctly.

---

> ### Author Response · Authors · 2025-11-25
> **Rebuttal 2**
>
> ## W1.2 Variable Memory Length
>
> Thank you for pointing this out. **We have updated the ablation study part in our in paper to discuss this issue deeply.**
>
> Indeed, selecting an appropriate memory size in the MemAgent setting involves certain trade-offs. A larger memory size allows the model to store more useful information, but it also introduces challenges in memory management and increases the likelihood of redundancy. Conversely, a smaller memory size may lead to insufficient storage capacity, causing key information to be lost and ultimately leaving the model without the necessary references when answering questions.
>
> To maintain a reasonable compression ratio while keeping the total context length within 8,192 tokens, we adopt the default configuration of memory size = 1024 and chunk size = 5000 based on preliminary validation results. As shown in the ablation study in Section 3.3.2, although different configurations yield slight variations in performance, **MemAgent remains robust** as long as the memory size is not overly constrained. **We also run an additional set of experiments** in which the sum of memory size and chunk size was kept constant; **MemAgent again showed stable performance, and in both experiment groups the default setting achieves the best results.**
>
> We also fully agree that **introducing more flexible memory-length mechanisms could be an important direction for future improvement**. In this regard, managing memory length may resemble controlling the "thinking length" of reasoning models: while a hard upper bound is necessary, one can still regulate output length using mechanisms such as the *thinking_effort* parameter. For memory systems, one possibility is to let the model receive signals about how much **capacity** has been used during memory updates. Another approach is to set an initial **soft limit** that can gradually increase as needed until reaching the hard limit—just like how `soft` and `hard` limit works in `ulimit` command in Unix systems.
>
> We appreciate your insight and view this as a valuable direction for future work.
>
> ## W2 Different task categories
>
> Thank you for asking this. We have added two classic long-document summarization benchmarks, **GovReport[1]** and **QMSum[2]**. The experimental results show that MemAgent also performs well on summarization tasks. We hope these results illustrate that the memory patterns learned during RL training naturally generalize to broader and more generic scenarios.
>
>
> > Comparison of RL-MemAgent and baseline models on summary benchmarks. All values represent recall rates (%). **Bold** = Highest value, *Italic* = Second highest value in each column.
>
> | Model                      | | |GOV REPORT | | | |QMSUM | |
> |----------------------------|-----------------|-----------------|-----------------|-----------------|-----------------|-----------------|-----------------|-----------------|
> |                            | ROUGE-1         | ROUGE-2         | ROUGE-L         | AVG             | ROUGE-1         | ROUGE-2         | ROUGE-L         | AVG             |
> | Qwen2.5-Instruct-32B       | 23.67           | 8.46            | 12.57           | 14.90           | 47.77           | 11.29           | 28.17           | 29.08           |
> | Qwen2.5-Instruct-14B       | 31.19           | 10.96           | 14.96           | 19.04           | 47.53           | 11.46           | 28.28           | 29.09           |
> | Qwen2.5-Instruct-7B        | 30.91           | 11.68           | 15.20           | 19.26           | 46.64           | 12.01           | 28.33           | 28.99           |
> | QwenLong-L1                | 27.60           | 8.20            | 13.07           | 16.29           | 39.44           | 8.24            | 23.56           | 23.74           |
> | Qwen2.5-Instruct-14B-1M    | 30.58           | 11.93           | 15.51           | *19.34*    | 47.31           | 13.13           | 29.07           | *29.84*    |
> | Qwen2.5-Instruct-7B-1M     | 31.02           | 11.47           | 15.30           | 19.26           | 46.72           | 12.33           | 28.66           | 29.24           |
> | DS-Distill-Qwen-32B        | 26.13           | 8.86            | 12.98           | 15.99           | 39.09           | 8.75            | 23.96           | 23.93           |
> | DS-Distill-Qwen-14B        | 28.24           | 9.72            | 13.78           | 17.25           | 41.25           | 8.95            | 25.00           | 25.07           |
> | DS-Distill-Qwen-7B         | 33.30           | 9.39            | 14.59           | 19.10           | 34.33           | 5.97            | 21.57           | 20.62           |
> | **RL-MemAgent-14B**       | **37.16**       | *12.03*    | **16.23**       | **21.80**       | **50.21**       | *12.70*    | **31.27**       | **31.39**       |
> | **RL-MemAgent-7B**        | 30.28           | **12.37**       | *15.37*    | *19.34*    | *48.49*    | **14.41**       | *30.91*    | *31.27*    |

---

> ### Author Response · Authors · 2025-11-25
> **Rebuttal 3**
>
> ## W3 & W4: Multi-Conv DAPO
>
> Thank you for your insights regarding RL Training. We genuinely hope to improve reproducibility and contribute further to the community, and we will **open-source our work**. Accordingly, **Appendix A** provides detailed descriptions of the data construction pipeline, hyperparameters, and training framework. Following your suggestion, we also added **pseudocode for Multi-Conv DAPO in Appendix A.2**. In recent months, several open-source training frameworks have begun to provide stronger support for agentic training; within these frameworks, one can initiate full training simply by referencing the prompt in Appendix A.1 and implementing the MemAgent loop.
>
> Although we did not implement finer-grained reward assignment due to engineering complexity, **we strongly agree with your point that distinguishing critical steps could meaningfully improve memory training**. For example, in the training set, the positions of key information within the context are predictable. After the model reads a key segment, we can extract its intermediate outputs to verify whether the relevant information has been correctly captured, and subsequently track whether this information is effectively preserved. We regard this as an important direction for future enhancement too, and we sincerely appreciate your insightful suggestion.
>
> ## Q1: Bridging the gap between graphical modeling and the RL formulation
>
> Thank you for your constructive question. **We have added a paragraph discussing this issue in Section 2.3 of the paper.**
>
> In our formulation, the model's operations over the context—reading and writing—constitute an MDP(Markov Decision Process). The goal of RL is to optimize the final reward obtained in this MDP. From the perspective of a language model, the generation process can be viewed as an alternating sequence of read–write operations that induces a probability distribution. Thus, RL is optimizing the policy model that governs these read–write actions.
>
> Therefore, MemAgent's learning objective is to generate a read–write memory trajectory that maximizes the reward, which corresponds to learning an optimal distribution over memory states conditioned on the input context.
>
> ## Reference
> > [1] Huang L, Cao S, Parulian N, Ji H, Wang L. Efficient attentions for long document summarization. arXiv preprint arXiv:2104.02112, 2021.
>
> > [2] Zhong M, Yin D, Yu T, Zaidi A, Mutuma M, Jha R, Hassan A, Celikyilmaz A, Liu Y, Qiu X, et al. QMSum: A new benchmark for query-based multi-domain meeting summarization. In: Proceedings of the 2021 Conference of the North American Chapter of the Association for Computational Linguistics: Human Language Technologies, 2021: 5905-5921.

---

### Official Review · Reviewer_FzMS · 2025-11-01

**Soundness:** 3
**Presentation:** 2
**Contribution:** 2
**Rating:** 4
**Confidence:** 3

**Summary:**

This paper presents MEMAGENT to enable LLMs to process arbitrarily long contexts efficiently by learning to maintain and overwrite a fixed-length memory.
Rather than increasing the model’s context window or modifying its architecture, MEMAGENT processes long documents in segments (chunks) and uses a learned memory update policy to selectively retain and discard information relevant to the final task.
Empirical results show that MEMAGENT significantly outperforms strong long-context baselines such as Qwen2.5-Instruct-1M, QwenLong-L1, and DeepSeek-Distill.

**Strengths:**

1. Memory update is formulated as a reinforcement learning setting.
2. The experiments across RULER-HQA, LongBench-QA, and NIAH show the effectiveness of the proposed method.
3. Detailed ablation study is conducted.

**Weaknesses:**

1. The comparison is weak. There are some other long-context modeling methods such as FocusLLM (FocusLLM: Precise Understanding of Long Context by Dynamic Condensing) and E2LLM (E2LLM: Encoder Elongated Large Language Models for Long-Context Understanding and Reasoning) which also use chunks. Moreover, there are also some memory-based methods such as Mem0. The proposed method should be compared with them.
2. Despite inference efficiency, the RL training process is computationally heavy. The time complexity of the proposed method is not discussed.
3. While the model overwrites memory continuously, the paper does not empirically analyze failure cases where crucial early information is overwritten, nor propose mitigation strategies.

**Questions:**

What is the comparison with other long-context modeling methods and memory-based methods?

---

> ### Author Response · Authors · 2025-11-25
> **Rebuttal 1**
>
> Dear Reviewer FzMS,
>
> Thank you for recognizing the novelty of using end-to-end RL to equip LLMs with memory capabilities, and for your thoughtful feedback. Below are our responses to the weaknesses and questions you raised:
>
> ## W1 & Q1: Comparison with other long-context modeling methods and memory-based methods
>
> Thank you for pointing this out. We first briefly discuss the similarities and differences between our work and the related methods you mentioned, followed by the latest experimental results completed during the rebuttal period. Our main conclusion is that **MemAgent offers broader applicability compared with chunk-based architectural methods, and demonstrates performance advantages over memory-based agents.**
>
> - **Long-context modeling methods.** The works you mentioned—FocusLLM [1] and E2LLM [2]—are excellent examples of applying the chunk strategy within the attention mechanism to enhance long-document modeling. While these methods introduce effective architectural improvements, MemAgent aims to achieve long-context capability *without modifying the model architecture or generation pattern*. This allows MemAgent to be post-trained on any backbone model using existing RL frameworks, and remain compatible with inference engines such as vLLM.
> In our **main experiments** shown in **Table1 and Table2**, we compared against the Qwen2.5-Instruct-1M series—obtained through CT and DCA extrapolation to 1M tokens—as well as QwenLong-L1, which is explicitly post-trained for long-context tasks. We hope these comparisons further strengthen your confidence in MemAgent relative to long-context modeling approaches.
>
> - **Memory agents.** Existing agent systems such as Mem0 [3] generally focus on chatbot agents, where the primary goal is to extract users’ personal experiences and preferences from conversational history. Although their fact-retrieval and memory-merging mechanisms are more sophisticated and carefully designed, they are oriented toward capturing fragmented, simple facts rather than effectively summarizing and organizing information from long documents. As a result, Mem0 is less suitable for general long-text reading tasks.
>
> To ensure a fair comparison, we implements Mem0 using SOTA OpenAI models, GPT-5.1 and text-embedding-3-large, and followed the [ official GitHub repository](https://github.com/mem0ai/mem0) for memory updating and retrieval. Specifically, during memory creation, we split and processed the entire context in 5,000-token chunks; during retrieval, we selected the top 30 memories.
>
> Although we have tuned the PE, Mem0 still fails to achieve strong performance—likely due to a mismatch between its intended application scenarios and our task. This does not diminish Mem0’s strong capabilities in chatbot settings, but highlights its limitations when applied to long-document understanding.
>
> | **Method**       | **2Wiki** | **HQA** | **MuSiQue** | **NQA** | **Qasper** | **RULER-HQA-7K** | **RULER-HQA-14K** | **RULER-HQA-28K** |
> |------------------|-----------|---------|-------------|---------|-----------|------------------|-------------------|-------------------|
> | RL-MemAgent-7B   | 74.00      | 69.50    | 47.00        | 21.50    | 29.00      | 81.25             | 81.25             | 82.03             |
> | Mem0             | 65.50     | 61.50   | 40.00       | 24.50   | 15.00     | 49.22            | 43.75             | 37.50             |
>
> The results reported in the Mem0 paper also show that RAG methods using only top-1 or top-2 retrieval already serve as strong and stable baselines for memory-agent tasks. RAG also constitutes a reasonable baseline for long-context tasks. Therefore, we conduct extensive comparisons with RAG agents that use text-embedding-3-large as the embedding model. **Experimental results demonstrate that RL-MemAgent consistently outperforms RAG agents across all configurations by a significant margin.**

---

> ### Author Response · Authors · 2025-11-25
> **Rebuttal 2**
>
> ## W1 & Q1: Experiments Results
>
> > Result versus RAG Agent in RULER-HQA with different top-K settings. We segment the context based on natural semantic units, i.e., each wiki item was treated as a chunk.
>
> | Model               | 7K     | 14K    | 28K    | 56K    | 112K   | 224K   | 448K   | 896K   |
> |---------------------|--------|--------|--------|--------|--------|--------|--------|--------|
> | RAG + Qwen2.5-14B   |        |        |        |        |        |        |        |        |
> | *K=2*               | 57.03  | 54.69  | 51.56  | 54.69  | 53.12  | 50.00  | 52.34  | 49.22  |
> | *K=4*               | 66.41  | 67.19  | 68.75  | 67.19  | 66.41  | 64.06  | 66.41  | 64.84  |
> | *K=6*               | 72.66  | 75.78  | 75.78  | 74.22  | 69.53  | 71.88  | 73.44  | 67.19  |
> | *K=8*               | 78.12 | 78.91 | 77.34 | 81.25 | 76.56 | 78.12 | 77.34 | 74.22 |
> | **RL-MemAgent-14B** | **80.47** | **82.03** | **82.03** | **83.59** | **81.25** | **77.34** | **79.69** | **75.78** |
> | RAG + Qwen2.5-7B    |        |        |        |        |        |        |        |        |
> | *K=2*               | 53.91  | 54.69  | 53.12  | 51.56  | 54.69  | 51.56  | 52.34  | 49.22  |
> | *K=4*               | 67.19  | 66.41  | 66.41  | 67.19  | 64.84  | 64.06  | 62.50  | 61.72  |
> | *K=6*               | 74.22  | 73.44  | 72.66  | 73.44  | 470.31 | 73.44  | 70.31  | 67.19  |
> | *K=8*               | 75.00 | 75.00 | 75.78 | 74.22 | 74.22 | 77.34 | 72.66 | 68.75 |
> | **RL-MemAgent-7B**  | **81.25** | **81.25** | **82.03** | **80.47** | **79.69** | **75.78** | **76.56** | **74.22** |
>
> > Result versus RAG Agent in Longbench-QA with different top-K and Context size settings. We segment the context using fixed-length chunks. For retrieval, we performed top-k matching using cosine similarity scores.
>
> | **Method**       | **2Wiki** | **HQA** | **MuSiQue** | **NQA** | **Qasper** | **AVG** |
> |----------------|--------|--------|--------|--------|--------|--------|
> |Qwen2.5-7B + RAG |   |   |   |   |   |   |
> |*C=1024 K=2* | 41.00 | 48.50 | 22.00 | 14.50 | 25.50 | 25.25 |
> |*C=1024 K=4* | 49.00 | 56.50 | 28.00 | 17.00 | 28.50 | 29.83 |
> |*C=1024 K=6* | 54.50 | 57.50 | 29.50 | 17.00 | 25.00 | 30.58 |
> |*C=1024 K=8* | 50.50 | 59.00 | 29.50 | 18.00 | 25.00 | 30.33 |
> |*C=2048 K=2* | 49.50 | 51.50 | 19.50 | 12.50 | 27.00 | 26.67 |
> |*C=2048 K=4* | 50.50 | 53.00 | 26.00 | 17.00 | 25.50 | 28.67 |
> |*C=2048 K=6* | 50.50 | 56.50 | 27.50 | **22.00** | 25.50 | 30.33 |
> |*C=2048 K=8* | 50.50 | 58.00 | 25.50 | 21.00 | 27.00 | 30.33 |
> |**RL-MemAgent-7B** | **74.00** | **69.50** | **47.00** | 21.50 | **29.00** | **48.20** |
> |Qwen2.5-14B + RAG |   |   |   |   |   |   |
> |*C=1024 K=2* | 51.50 | 56.50 | 26.50 | 15.00 | 23.50 | 28.83 |
> |*C=1024 K=4* | 70.00 | 64.50 | 34.50 | 17.50 | 27.00 | 35.58 |
> |*C=1024 K=6* | 71.50 | 64.00 | 41.00 | 19.00 | **27.50** | 37.17 |
> |*C=1024 K=8* | 72.50 | 64.50 | 39.00 | 17.50 | 26.00 | 36.58 |
> |*C=2048 K=2* | 58.50 | 61.50 | 33.50 | 13.50 | 25.50 | 32.08 |
> |*C=2048 K=4* | 76.00 | 64.00 | 36.00 | 18.50 | 25.00 | 36.58 |
> |*C=2048 K=6* | 73.00 | 67.50 | 41.50 | 21.00 | 26.50 | 38.25 |
> |*C=2048 K=8* | 77.50 | 68.50 | 42.00 | 21.00 | **27.50** | 39.42 |
> |**RL-MemAgent-14B** | **79.0** | **73.0** | **52.00** | **25.00** | 26.00 | **51.00** |

---

> ### Author Response · Authors · 2025-11-25
> **Rebuttal 3**
>
> ## W2: Computation Complexity
>
> Thank you for raising this point. We agree that computational efficiency is crucial in RL training. To address this concern, we estimated the FLOPs of MemAgent-7B and the Qwen2.5-7B-Instruct-1M model. **MemAgent benefits from an $O(n)$ computational complexity due to its fixed-length chunk strategy, whereas naïve long-context models incur an $O(n^2)$ cost**.
>
> In addition, because MemAgent’s rollout process consists of multiple short segments with fixed length, its **peak GPU memory usage is substantially lower** than that of directly training a long-context model. In practice, this **enables better parallelism (e.g., larger batch sizes), which in turn leads to faster training.**
>
> > Computation required acrossing different context length in TFLOPS.
>
> |Method | 8K | 16K | 32K | 64K | 128K | 256K | 512K |
> |---------------|----|-----|-----|-----|------|------|------|
> |MemAgent-7B|      782.34  | 1468.85 |  2729.80 |  5367.9   | 10524.0   | 20838.26   | 41475.08 |
> |Qwen2.5-7B-Instruct-1M| 601.20   | 1260.33 |  3063.47 |  8609.3   | 27459.1   | 96191.31   | 357786.21 |
>
> ## W3: Failure Pattern
>
> Thank you for raising this question. Several reviewers mentioned this concern, and we treated it very seriously by conducting a detailed analysis.
>
> 1. We added **a more comprehensive set of failure case studies** from the 3.5M-length test set which are available in **Appendix F**. These cases illustrate concrete patterns in which key information is forgotten early or not properly identified as important.
>
> 2. Our hypothesis is that **overcoming the information-overwritten problem is a natural result of end-to-end optimization.** During training, the model learns to preserve and track critical information in order to maximize the final reward.
>
> 3. To validate this hypothesis, we carefully design **a set of probing experiments for RULER-HQA**. **The probing experiment results show that MemAgent remains consistently robust across all patterns without exhibiting any catastrophic performance degradation.**
>
> We divided the key information into two groups and placed them at different positions within the context. We constructed five settings: (0%, 100%), (20%, 80%), (40%, 60%), (0%, 20%), and (80%, 100%).
>    - In the (0%, 100%) case, the model sees one piece of key information at the very beginning and the other only at the final memory update step. This represents one of the most challenging scenarios for the information-overwritten problem.
>
>
> > **Probing experiment results.** “random’’ indicates randomly shuffling all context items, consistent with the setup in the main experiment. The other rows show the performance difference relative to “random.’’ “Ctx. Dist.’’ denotes the context distribution, where the two numbers correspond to the relative positions of the two key-information groups within the entire context. 0% means the beginning and 100% means the end.
>
> | Model | Ctx. Dist.   | 7K      | 14K       | 28K       | 56K       | 112K      | 224K      | 448K       | AVG       |
> |-------|--------------|---------|-----------|-----------|-----------|-----------|-----------|------------|-----------|
> | 14B   | random       | +80.47  | +82.03    | +82.03    | +83.59    | +81.25    | +77.34    | +79.69     | +75.78    |
> |       | 0% 20%       | +3.91   | -3.91     | +3.13     | +1.57     | 0.00      | +4.69     | +3.90      | +1.90     |
> |       | 0% 100%      | +3.12   | +0.78     | +3.13     | -3.12     | +1.56     | +7.82     | +6.25      | +2.79     |
> |       | 20% 80%      | +0.78   | -3.12     | +2.35     | +0.79     | -3.13     | +5.47     | -3.13      | 0.00      |
> |       | 40% 60%      | +1.56   | +2.35     | -2.34     | -3.12     | -1.56     | +3.13     | -0.78      | -0.11     |
> |       | 80% 100%     | -2.35   | 0.00      | +1.56     | +0.79     | +3.13     | 0.00      | +1.56      | +0.67     |
> | 7B    | random       | +81.25  | +81.25    | +82.03    | +80.47    | +79.69    | +75.78    | +76.56     | +79.58    |
> |       | 0% 20%       | -1.56   | -0.78     | +3.13     | +3.91     | +3.12     | +5.47     | +3.13      | +2.35     |
> |       | 0% 100%      | +0.78   | 0.00      | +2.35     | +1.56     | +3.90     | +4.69     | +2.35      | +2.23     |
> |       | 20% 80%      | -0.78   | -0.78     | +2.35     | 0.00      | 0.00      | 0.00      | +3.13      | +0.56     |
> |       | 40% 60%      | 0.00    | +1.56     | +3.13     | +0.78     | -3.91     | +3.13     | +5.47      | +1.45     |
> |       | 80% 100%     | +1.56   | 0.00      | +0.78     | -0.78     | +0.78     | +4.69     | 0.00       | +1.00     |

---

> ### Author Response · Authors · 2025-11-25
> **Rebuttal 4**
>
> ## W3: Failure Pattern
>
> **For case study, we manually check the memory changing and highlight the core difference, and write a short summary for each sample.  The full cases are available in Appendix F**. We hope these analyses offer further intuitive insight into how MemAgent behaves in challenging scenarios.
> - **Failure in Writing: Information Overwritten**: In this example, the model accumulates a large amount of irrelevant memories in the early stage (Turn 60). When the crucial context appears, the model does capture the relevant information (Turn 289), but attempts to append it to the end (Turn 290), where it is truncated due to insufficient memory. Later, the model proactively performs a summarization (Turn 317), which gives it more space to store the second key piece of context (Turn 433). However, since the information from the first context has already been lost, the model incorrectly relies on irrelevant information for reasoning and fails to answer the question correctly.
> - **Failure in Reading: Missing Key Information**: In this example, when the model encounters the first key piece of information (Turn 143), it fails to recognize it as such because this is a multi-hop question and the prerequisite evidence has not yet been observed. Consequently, even though the model becomes aware of the problem upon seeing the second piece of information (Turn 215), it still fails to produce the correct final answer.
> - **Failure in Reasoning: Primacy Bias** The word *country* can refer either to a nation or to the countryside. The model initially assumes it refers to a nation (Turn 1) and subsequently reinforces this belief when it encounters additional information about countries (Turn 617). As a result, even though it eventually also attends to the truly critical piece of information (Turn 728), it still fails to answer the question correctly.
>
> > [1] Li Z, Zhang Y, Pan T, Sun Y, Duan Z, Fang J, Han R, Wang Z, Wang J. Focusllm: Precise understanding of long context by dynamic condensing. In: Proceedings of the 63rd Annual Meeting of the Association for Computational Linguistics (Volume 1: Long Papers), 2025: 31087-31101.
>
> > [2] Liao Z, Wang J, Yu H, Wei L, Li J, Zhang W. E2llm: Encoder elongated large language models for long-context understanding and reasoning. In: Proceedings of the 2025 Conference on Empirical Methods in Natural Language Processing, 2025: 19212-19241.
>
> > [3] Chhikara P, Khant D, Aryan S, Singh T, Yadav D. Mem0: Building production-ready ai agents with scalable long-term memory. arXiv preprint arXiv:2504.19413, 2025.

---

### Author Response · Authors · 2025-11-26
**General Response 1**

We sincerely thank all the reviewers for carefully reading our paper and for your constructive feedback and comments. We take your suggestions very seriously and have added additional experiments and discussions to address the raised concerns.

We have provided point-by-point responses to all comments and updated the manuscript accordingly. **The revised parts are highlighted in blue** (and in red for the Related Work section so that they stand out from blue hyperlinks).

The changelist in this revision are summarized as follows:

1. **Probing Experiments** over different contect distribution(§3.3.3 P9-10) and **Failure Pattern Analysis**(§F P24-29)
> FzMS, BjEW, BZGs and JGDV
2. **Summary Benchmark**: GovReport and QMSum  (§3.1 §3.2 P6-8)
> BjEW, BZGs and JGDV
3. **Agent Baselines**: Mem0 and RAG (§D.2 P22-23)
> FzMS, JGDV
4. **Memory Size**: further discussion and more ablation(§3.3.2 P8 & §D.1 P22)
> BjEW, BZGs

> FzMS
1. **Computation Complexity** (§B P19)
2. **Comprison with Long-context Modeling Method** (§4.1 P10)

> BjEW
1. **Algorithm Persudo Code** (§ A.2 P17)
2. **Formulation** (§ 2.3 P6)

We once again thank you for helping us polish this work, and look forward for your further feedback

---

### Author Response · Authors · 2025-12-03
**Summary 1**

Dear AC and SAC,

We appreciate the time and effort you have dedicated to reviewing this paper! We also thank all the reviewers for their time and insightful feedback. Considering the heavy workload caused by the special arrangements of ICLR26, we have written a summary to highlight the main advantages of the paper, the reviewers' concerns, and our responses.

The reviewers recognize MemAgent as a **novel and efficient agent-based long-text modeling approach** (BjEW, JGDV), with **effective utilization of the RL framework** (FzMS, BZGs), demonstrating **excellent performance in long-text tasks** (FzMS, BjEW, BZGs, JGDV). We also appreciate the reviewers' encouragement regarding our **writing** (BZGs) and **ablation studies** (FzMS).

During the Rebuttal phase, **we addressed all the reviewers' concerns, provided detailed discussions, and conducted extensive additional experiments to validate our results.** One of the reviewers (BZGs) acknowledged our **detailed reply and extensive additional results** and raised further questions, all of which we have responded to.

## Discussion with Reviewer FzMS
>**Q1 & W1**: Lack of baselines using long-text models and memory-based agents

- (**Table 1, 2, 3, Figure 5**) In the main experiments, MemAgent **consistently outperforms** Qwen-1M and QwenLong-L1 series, which have been trained on long texts.
- (**§4.1 P10**) We analyze that MemAgent provides a way to use **a general LLM** to acquire memory abilities through post-training, while long-text methods based on model architecture lack **compatibility**.
- (**§D.2 P22-23**) We compare MemAgent with Mem0 and RAG in **Exp3: Agent Baselines**, where MemAgent exhibits **consistently excellent performance**.

> **Q2**: RL training process is computationally heavy

- (**§B P19**) We show that MemAgent has an **better $O(n)$ time complexity** compared to traditional methods, which has been acknowledged by the reviewers (BjEW, JGDV).
- We further analyze that, in practice, MemAgent has **lower peak memory usage, which further accelerates training.**

>  **Q3**: Lack of failure pattern analysis, concerns over early memory coverage

- (**§F P24-29**) We provide a **detailed analysis of three failure patterns**: Failure in Writing (Information Overwritten), Failure in Reading (Missing Key Information), and Failure in Reasoning (Primacy Bias). We also offer **specific explanations** for the causes of errors and the **diff visualization** of memory changes.
- (**§3.3.3 P9-10**) In **Exp1: Probing Experiments**, we demonstrate that **even when some useful information appears at the beginning and other useful information at the end, MemAgent still achieves the same excellent performance**, proving that early memory overwritten is naturally addressed through RL learning.

## Discussion with Reviewer BjEW
>  **W1**: Concerns about fixed-memory size

- (**§3.3.2 P8**) We explain that **the initial hyperparameters** of MemAgent are chosen after considering **both the context window and compression ratio**, which has been **experimentally verified as optimal**. Additionally, **changes in memory size have a minor impact on MemAgent's performance**. Even with an extreme compression of memory to 256 tokens, MemAgent still maintains **robust performance**.
- (**§D.1 P22**) In **Exp4: Memory Size**, we add **more ablation experiments** on memory size and context size to **show MemAgent's robustness**.
- The reviewer suggests future improvements, including variable memory size. We appreciate this idea and propose several heuristic methods to implement it. However, based on the experiments mentioned above, we believe that limiting the memory upper bound is sufficient for the model, as it efficiently utilizes memory and discards irrelevant information.

> **W2**: Single task type

- (**§3.1 §3.2 P6-8**) In **Exp2: Summary Benchmark**, we add experiments on the new task type, Longsummary, which shows that MemAgent still leads the results, proving that the learned memory capabilities are not task-specific.

> **W3**: Concerns about reproducibility

- (**§ A.2 P17**) We **commit to open-sourcing the code** and, **upon the reviewer's request, we have added pseudocode** in **§ A.2**. Additionally, we have written a **reproducibility statement** and provided **detailed implementation specifics** in **§ A**.

> **W4**: Suggestions for improving reward assignment

- The current reward assignment is **already effective** in training MemAgent and achieving excellent results. We appreciate the reviewer's idea and discuss some finer methods as future directions.

> **Q1**: Questions about the relationship between formal modeling and RL objective function

- (**§2.3 P6**) We respond to the reviewer and add an explanation to the paper. The memory read/write process constitutes a Markov process, and the RL optimization objective is to maximize the reward of this process.

---

### Author Response · Authors · 2025-12-03
**Summary 2**

## Discussion with Reviewer BZGs
>  **W1**: Memory management may be a problem & **W2**: Memory length may be costly to tune

- Similar to **Reviewer BjEW W1**, we discuss in (**§3.3.2 P8**) and (**§D.1 P22**) in **Exp4: Memory Size** that MemAgent is **robust to changes in memory size and context size**. Adjusting the memory size **does not cause significant memory management issues**, and thus **no additional hyperparameter tuning is necessary.**

>  **Q1**: Can MemAgent be adapted to other tasks, like long-form summarization?

- (**§3.1 §3.2 P6-8**)  In **Exp2: Summary Benchmark**, we demonstrate that MemAgent **generalizes to summarization tasks**.

>  **Q1 Round2**: Concerns about reward hacking

- We address the concern about the model possibly *hacking the ROUGE score by simply repeating the original text*. In our response, we provide results of the longest common substring detection, proving that **the model does not exhibit this tendency**.

> **Q2**: Have you explored the lost-in-the-middle for MemAgent?

- (**§3.3.3 P9-10**) In **Exp1: Probing Experiments**, we find that MemAgent is **almost unaffected by the "lost-in-the-middle" phenomenon**, with the maximum average performance drop only being 0.11%.

>  **Q2 Round2**: Is RL-MemAgent robust because of the unbiased training data or due to architectural design forcing it to keep good information in memory?

- We analyze the influence of these two factors and point out that unbiased data is a necessary condition, while the architecture further strengthens the model's ability to manage memory effectively.

## Discussion with Reviewer JGDV
> **Q1 & W2**: Could you provide a failure mode analysis, especially in the 3.5M token QA task? Does the distribution of errors relate to the distance of required context?

- (**§F P24-29**) Similar to **Reviewer FzMS Q3**, we explain the **three failure patterns on the 3.5M-length task and provide a detailed analysis.**

- (**§3.3.3 P9-10**) In **Exp1: Probing Experiments**, we quantitatively explore the impact of context distribution on performance. MemAgent shows **consistent robustness** and is **more sensitive to information at the beginning and end.**

>  **Q2 & W1**: Could you provide results on a high-quality long-document summarization benchmark?

- (**§3.1 §3.2 P6-8**) Similar to **Reviewer BjEW W2**, we conduct **Exp2: Summary Benchmark** to demonstrate MemAgent's **generalization ability**.

> **Q3 & W3**: Could you discuss the performance comparison with other agent solutions as well?

- (**§D.2 P22-23**) Similar to **Reviewer FzMS Q1 & W1**, we compare MemAgent with Mem0 and RAG in **Exp3: Agent Baselines**, showing MemAgent's **superior performance over these baselines**.

Overall, we have conducted four additional experiments and addressed a substantial number of concerns raised by the reviewers. We hope this makes our paper more comprehensive and that this summary proves helpful to you.

- Exp1: Probing Experiments over different contect distribution(§3.3.3 P9-10) and Failure Pattern Analysis(§F P24-29)
- Exp2: Summary Benchmark: GovReport and QMSum (§3.1 §3.2 P6-8)
- Exp3: Agent Baselines: Mem0 and RAG (§D.2 P22-23)
- Exp4: Memory Size: further discussion and more ablation(§3.3.2 P8 & §D.1 P22)

Sincerely,
The Authors

---

### Meta-Review · Area_Chair_xpBH · 2026-01-07

**Summary:**

This paper proposes an agent-based post-training framework based on RL to enable LLMs to handle long-context inputs. The approach processes long documents in chunks and uses RL post-training to learn to update a fixed-size memory across chunks, thereby achieving linear-time inference with respect to input length.

Several common concerns were raised by the reviewers. These include the lack of comparisons with other agent-based or long-context methods (FzMS, JGDV), limited task diversity with most evaluations focusing on extractive tasks and missing other tasks like long-form summarization (BjEW, BZGs, JGDV), the absence of failure mode analysis (FzMS, JGDV), and questions regarding the effect and tuning of memory size (BjEW, BZGs). The authors have responded thoroughly to these concerns by providing additional experiments, expanded analyses, and clearer discussions in the revised manuscript.

Overall, the proposed approach demonstrates strong effectiveness and efficiency for long-text modeling, and the rebuttal substantially strengthens the paper. I therefore recommend acceptance.

**Reviewer Concerns:**

Addressed Concerns:
- Comparisons with other agent-based or long-context methods: The authors provide comparisons with both long-context models and agent-based approaches designed for long-context problems.
- Limited task diversity: The authors add evaluations on two long-form summarization tasks from LongBench.
- Absence of failure mode analysis: The authors present an analysis of three failure patterns and discuss the impact of context distribution.
- Effect and tuning of memory size: The authors include additional discussions on memory size and context length.

Outstanding Concerns:
- Computational cost of RL training: While the authors argue that the RL rollout process has linear complexity and is more efficient than long-context models for training, the rebuttal does not provide a direct or quantitative comparison of the full RL training cost against other agent-based long-context methods.

**Reviewer Scores:**

For reviewers who leaned toward acceptance, I expect they would likely maintain or slightly increase their scores, as the rebuttal and additional experiments addressed their key concerns and further strengthened the paper.

For reviewers who leaned toward rejection, the major concerns regarding the lack of baselines and failure mode analysis were addressed through additional experiments and detailed analyses. While concerns about the computational cost of RL training remain partially outstanding, the overall response improves the paper and may lead to a more favorable score.

---

### Decision · Program_Chairs · 2026-01-26

Accept (Oral)